# Ozone Pollution over China and India: Seasonality and Sources

Meng Gao[1,2,3], Jinhui Gao[4], Bin Zhu[3], Rajesh Kumar[5], Xiao Lu[2], Shaojie Song[2], Yuzhong Zhang[2], Beixi Jia[1], Peng Wang[6], Gufran Beig[7], Jianlin Hu[8], Qi Ying[6], Hongliang Zhang[9], Peter Sherman[10], and Michael B McElroy[2,10]

Department of Geography, State Key Laboratory of Environmental and Biological Analysis, Hong Kong Baptist University, Hong Kong SAR, China
John A. Paulson School of Engineering and Applied Sciences, Harvard University, Cambridge, MA, United States
Key Laboratory for Aerosol-Cloud-Precipitation of China Meteorological Administration, Nanjing University of Information Science and Technology, Nanjing, China
Department of Ocean Science and Engineering, Southern University of Science and Technology, Shenzhen, China
National Center for Atmospheric Research, Boulder, CO, USA
Zachry Department of Civil Engineering, Texas A&M University, College Station, TX 77843-3136, USA
Indian Institute of Tropical Meteorology, Pune, India
School of Environmental Science and Engineering, Nanjing University of Information Science & Technology, 219 Ningliu Road, Nanjing 210044, China
Department of Environmental Science and Engineering, Fudan University, Shanghai 200438, China
Department of Earth and Planetary Sciences, Harvard University, Cambridge, MA, United States

**Correspondence:** Meng Gao (mmgao2@hkbu.edu.hk) and Michael B. McElroy (mbm@seas.harvard.edu)

**Abstract**

A regional fully coupled meteorology-chemistry Weather Research and Forecasting model with Chemistry (WRF-Chem) was employed to study the seasonality of ozone ($O_3$) pollution and its sources in both China and India. Observations and model results suggest that $O_3$ in the North China Plain (NCP), Yangtze River Delta (YRD), Pearl River Delta (PRD) and India exhibit distinctive seasonal features, which are linked to the influence of summer monsoons. Through a factor separation approach, we examined the sensitivity of $O_3$ to individual anthropogenic, biogenic, and biomass burning emissions. We found that summer $O_3$ formation in China is more sensitive to industrial and biogenic sources than to other source sectors, while the transportation and biogenic sources are more important in all seasons for India. Tagged simulations suggest that local sources play an important role in the formation of the summer $O_3$ peak in the NCP, but sources from Northwest China should not be neglected to control summer $O_3$ in the NCP. For the YRD region, prevailing winds and cleaner air from the ocean in summer lead to reduced transport from polluted regions, and the major source region in addition to local sources is Southeast China. For the PRD region, the upwind region is replaced by contributions from polluted PRD as autumn approaches, leading to an autumn peak. The major upwind regions in autumn for the PRD are YRD (11%) and Southeast China (10%). For India, sources in North India are more important than sources in the south. These analyses emphasize the relative importance of source sectors and regions as they change with seasons, providing important implications for $O_3$ control strategies.

**1 Introduction**

Tropospheric ozone ($O_3$) is the third most potent greenhouse gas in the atmosphere (*Pachauri and Reisinger, 2007*), an important surface air pollutant, and the major source of the hydroxyl radical (a key oxidant playing an essential role in atmospheric chemistry). With the rapid growth of industrialization, urbanization and transportation activities, emissions of $O_3$ precursors (nitrogen oxides and volatile organic compounds) in both China and India have increased significantly since 2000 (*De Smedt et al., 2010; Duncan et al., 2014; Hilboll et al., 2013; Kurokawa et al., 2013; Ohara et al., 2007; Stavrakou et al., 2009; Zheng et al., 2018*).

Increasing concentrations of $O_3$ precursors have led to emerging and widespread $O_3$ pollution,
threatening health and food security (*Chameides et al., 1994; Malley et al., 2018*). The decrease
in crop yield resulting from the increase in surface $O_3$ would have been sufficient to feed 95
million people in India (*Ghude et al., 2014*).
Great efforts have been devoted to improving understanding of exceptionally high
concentrations (*Wang et al., 2006*) and the increasing trend in $O_3$ for both China and India (*Beig*
*et al., 2007; Cheng et al., 2016; Ghude et al., 2008; Lu et al., 2018a; Ma et al., 2016; Saraf*
*and Beig, 2004; Xu et al., 2008*). Strong but distinctive seasonal variations of $O_3$ observed in
India and China have been linked to higher emissions of precursor gases (*Lal et al., 2000*), and
summer monsoon (*Kumar et al., 2010; Lu et al., 2018b; Wang et al., 2017*). The contributions
of individual economic sectors and source regions were reported based on sensitivity
simulations and source apportionment techniques (*Gao et al., 2016a; Li et al., 2008; Li et al.,*
*2016; Li et al., 2012; Lu et al., 2019; Wang et al., 2019*). With respect to the enhanced
concentrations of $O_3$ over the past years, *Sun et al. (2019)* attributed this to elevated emissions
of anthropogenic VOCs, while *Li et al. (2019)* argued that an inhibited aerosol sink for
hydroperoxyl radicals induced by decreased $PM_{2.5}$ over 2013-2017 played a more important
role in the NCP.
Despite these progresses, the seasonal behaviors of $O_3$ in different regions greatly differ, yet
have not been intercompared and the underlying causes have not been comprehensively
explored. In addition, previous source apportionment studies focused on specific regions or
episodes, and the policy implications drawn from these studies might not be applicable for
other regions and seasons. It is both of interest and of significance to understand the similarities
and differences between $O_3$ pollution in China and India, the two most polluted and most
populous countries in the world.
The present study uses a fully online coupled meteorology-chemistry model (WRF-Chem) to
examine the general seasonal features of $O_3$ pollution, and its sources derived from economic
sectors and regions over both China and India. Sect. 2 describes the air quality model and
measurements. We examine then in Sect. 3 how the model captures the spatial and temporal
variations of $O_3$ and relevant precursors. Sect. 4 presents general seasonal features of $O_3$
pollution, and the relative importance of both economic sectors and source regions. Results are
discussed and summarized in Sect. 5.

**2 Model and data**
**2.1 WRF-Chem model and configurations**
The fully online coupled meteorology-chemistry model WRF-Chem (*Grell et al., 2005*) was
employed in this study using the CBMZ (Carbon Bond Mechanism version Z, *Zaveri and*
*Peters, 1999*) photochemical mechanism and the MOSAIC (Model for simulating aerosol
interactions and chemistry, *Zaveri et al., 2008*) aerosol chemistry module. The model was
configured with a horizontal grid spacing of 60km with 27 vertical layers (from the surface to
10 hPa), covering East and South Asia (Fig. 1). The selected physical parameterization schemes
follow the settings documented in *Gao et al. (2016b)*, and they are listed in Table S1.
Meteorological initial and boundary conditions were obtained from the 6-hourly FNL (final
analyses, *NCEP, 2000*) global analysis data with $1.0°×1.0°$ resolution. The four-dimensional
data assimilation (FDDA) technique was applied to limit errors in simulated meteorology.
Horizontal winds, temperature and moisture were nudged at all vertical levels. Chemical initial
and boundary conditions were provided using MOZART-4 (*Emmons et al., 2010*) global
simulations of chemical species.
Monthly anthropogenic emissions of $SO_2$, $NO_x$, CO, NMVOCs (Non-methane Volatile Organic
Compounds), $NH_3$, $PM_{2.5}$, $PM_{10}$, BC (black carbon) and OC (organic carbon) were taken from
the MIX 2010 inventory (*Li et al., 2017*), a mosaic Asian anthropogenic emission inventory
covering both China and India. In this study, the emissions in China were updated with the
MEIC (Multi-resolution Emission Inventory for China, http://www.meicmodel.org/) inventory
for year 2012. From 2012 to 2013, emissions of $SO_2$ and $NO_x$ in China declined by 11% and
5%, while emissions of other species did not exhibit a significant change (*Zheng et al., 2018*).
The MIX inventory was prepared considering five economic sectors on a $0.25°×0.25°$ grid:
power, industrial, residential (heating, combustion, solvent use, and waste disposal),
transportation and agriculture. For India, $SO_2$, BC, OC, and power plant $NO_x$ emissions were
taken from the inventory developed by the Argonne National Laboratory (ANL), with the
REAS (Regional Emission inventory in Asia) inventory used to supplement for missing species.
Speciation mapping of VOCs emissions follows the speciation framework documented in *Li et*
*al. (2014)* and *Gao et al. (2018)*. The MEGAN (Model for Emissions of Gases and Aerosols
from Nature, *Guenther et al., 2012*) model version 2.04 was used to generate biogenic
emissions online. Biomass burning emissions were obtained from the 4[th] generation global fire
emissions database (GFED4, *Giglio et al., 2013*). For China, industrial and power sectors are
the largest two contributors to $NO_x$ emissions, while industrial sector emits the largest amounts
of NMVOCs (*Li et al., 2017*). For India, transportation and power sectors produce the largest
amounts of $NO_x$, while residential and transportation sectors are the largest two contributors to
NMVOCs emissions (*Li et al., 2017*). China's biogenic emissions of VOCs are estimated to be
comparable to or higher than anthropogenic sources (*Li and Xie, 2014; Wei et al., 2011*).

**2.2 Ozone tagging method and setting of source regions**
$O_3$ observed in a particular region is a mixture of $O_3$ formed by reactions of $NO_x$ with VOCs
emitted at different locations and time. The $O_3$ tagging method has the capability to apportion
contributions of different source regions to $O_3$ concentrations observed in particular regions.
The present study adopted the ozone tagging method implemented in WRF-Chem by *Gao et*
*al. (2017a)*, which is similar to the Ozone Source Apportionment Technology (OSAT, *Yarwood*
*et al., 1996*) approach implemented in the Comprehensive Air Quality Model with extensions
(CAMx). Both $O_3$ and its precursors from different source regions are tracked as independent
variables. The ratio of formaldehyde to reactive nitrogen oxides ($HCHO/NO_y$) was used as
proposed by *Sillman (1995)* to decide whether the grid cell is under $NO_x$ or VOC limited
conditions, and then different equations for these two conditions were selected to calculate total
$O_3$ chemical production. A detailed description of the technique is provided in *Gao et al.*
*(2017a)*.
The $O_3$ tagging method attributes production of $O_3$ and its precursors to individual geographic
areas. We divided the entire modeling domain into 23 source regions, which were classified
mainly using the administrative boundaries of provinces. In eastern China, each province was
considered as a source region, while provinces in northeastern, northwestern, and southwestern
China were lumped together (Fig. S1). India was divided into two source regions (north and
south), and other countries were considered separately as a whole (Fig. S1). Additionally, the
chemical boundaries provided by MOZART-4 were adopted to specify inputs of $O_3$, and the
initial condition was tracked also as an independent source. The names of all source groupings
are indicated in Fig. S1.

**2.3 Experiment design**

To quantify the sectoral contributions to $O_3$, a factor separation approach (FSA) was applied to
differentiate two model simulations: one with all emission sources considered, and the other
with some emission sources excluded. Table 1 summarizes the different sets of simulations
conducted in this study. In addition to the control case, a series of sensitivity studies was
performed, in which industrial, residential, transport, power, biogenic and fire emissions were
separately excluded (Table 1). For each case, the entire year of 2013 was simulated.

**2.4 Measurements**

Surface air pollutants in China are measured and recorded by the Ministry of Environmental
Protection (MEP), and the data are accessible on the China National Environmental Monitoring
Center (CNEMC) website (http://106.37.208.233:20035/). This nationwide network was
initiated in January 2013, and this dataset was used to evaluate model performance. This dataset
has been extensively employed in previous studies to understand the spatial and temporal
variations of air pollution in China (*Hu et al., 2016; Lu et al., 2018a*), and to reduce
uncertainties in estimates of health and climate effects (*Gao et al., 2017b*). Measurements of
air pollutants from the MAPAN network (Modeling of Atmospheric Pollution and Networking)
set up by the Indian Institute of Tropical Meteorology (IITM) under project SAFAR (System
of Air Quality and weather Forecasting And Research) (*Beig et al., 2015*) were used in the
present study to evaluate the model performance over India. To further evaluate how the model
performed in capturing the vertical distributions of $O_3$, we used data from ozonesonde records
obtained from the World Ozone and Ultraviolet Radiation Data Center website
(https://woudc.org/data/dataset_info.php?id=ozonesonde). Fig. 1 displays the locations of the
relevant surface and ozonesonde observation sites. We evaluated also the spatial distribution of
$NO_2$ columns using the KNMI-DOMINO (Dutch OMI $NO_2$) daily level-2 products of
tropospheric $NO_2$ column (www.temis.nl), with row anomaly removed (according to
operational flagging), solar zenith angles less than 80º, and cloud fraction less than 0.2. The
model results were sampled according to selected satellite data on a pair-to-pair basis. The
matched model results were transformed by applying the OMI averaging kernel to the modeled
vertical profiles of $NO_2$ concentrations.

**3 Model evaluation**
We evaluated the spatial distribution of simulated seasonal mean (winter months include
January, February and December (DJF); spring months include March, April and May (MAM);
summer months include June, July and August (JJA); Autumn months include September,
October, and November (SON)) $O_3$ concentrations by comparing model results with
observations (filled circles in Fig. 2) for 62 cities in China and India. The model captures the
spatiotemporal patterns of $O_3$ in east China, with lower values in fall (Fig. 2d) and winter (Fig.
2a), and enhanced levels in spring (Fig. 2b) and summer (Fig. 2c). However, $O_3$ concentrations
are overestimated by the model in central, northwest and southwest China for all seasons (Fig.
2). *Hu et al. (2016)* reported also that their model tends to predict higher $O_3$ concentrations for
these regions. Scatter plots of simulated and observed $O_3$ for four seasons suggest that model
overestimates $O_3$ in most sites during winter, and exhibit better performance during other
seasons (Fig. 3). Fig. S2 indicates that modeled $NO_2$ column values in east China are not as
high as observed, but model overpredicts $NO_2$ column in central China and most parts of India,
which could partly explain the overestimation of $O_3$ in central China.
We conducted a further site-by-site evaluation of monthly mean $O_3$ concentrations, and we
grouped stations into four major densely-populated regions, namely North China Plain (NCP),
Yangtze River Delta (YRD), Pearl River Delta (PRD), and India. The seasonality of observed
$O_3$ concentrations is reproduced well in these four regions (Fig. 4), although concentrations are
underestimated in the NCP in spring. $O_3$ concentrations in October, November and December
in the PRD region are overestimated by the model. The correlation coefficients between model
and observations range between 0.84 and 0.98. Detailed model evaluation statistics are
documented in Table 2. In Beijing, the daily maximum 8-h average (MDA8) $O_3$ concentrations
are well captured by the model (Fig. S3), except that the model is biased low in spring. Stronger
$NO_x$ titration (underestimation of $O_3$ during the night, Fig. S4) are found in the model results
for the NCP and YRD regions in spring, which can partly explain the underestimation of $O_3$ in

spring in the NCP and PRD (Fig. S4). The simulated magnitudes of $O_3$ in India are generally consistent with observations, though lower in central India and in May. The high concentrations of $O_3$ in India were not captured by the model is mainly because of the large underestimation in Jabalpur (Central India) with complex terrain. Model's coarse resolution and poor capability of resolving strong spatial heterogeneity in land types within a small area have led to this mismatch, which was also found in *Sharma et al. (2017)*. Fig. 4 suggests also that the seasonal behavior of $O_3$ in these four major regions exhibits distinctive patterns, discussed in detail in Sect. 4.

In this work, ozonesonde measurements from the Hong Kong Observatory (HKO), Japan Meteorological Agency (JMA), and the Hydrometeorological Service of S.R. Vietnam (HSSRV) (locations marked in purple in Fig. 1) were used. Wintertime near-surface $O_3$ concentrations are overestimated for HKO (Fig. S5), while vertical variations are satisfactorily captured by the model. Comparisons of near-surface $O_3$ precursors suggest that CO concentrations are underestimated in all the regions (Fig. S6), which could be explained by an underestimate of CO emissions (*Wang et al., 2011*). The coarse grid resolution of the model might provide another reason for this underestimation, as the observation sites in China are located mostly in urban areas. Underestimates of CO concentrations are reported also for many sites in India (*Hakim et al., 2019*). The effects of underestimated CO on $O_3$ were found to be small, but the underestimation of CO may lead to bias in methane lifetime (*Strode et al., 2015*), which is beyond the discussion of regional pollution in this study. Simulated $NO_2$ concentrations are slightly overestimated in the NCP but are underestimated in the PRD (Fig. S6). Despite these issues, the model still captures the seasonal behavior of $O_3$ in different regions, and we do not expect the model biases to change the major findings of the present study.

**4 Seasonality, source sectors and source regions**

**4.1 Seasonality of surface $O_3$ in different regions**

Comparisons between modeled and observed near-surface $O_3$ concentrations for different regions suggest distinctive seasonal patterns (Fig. 4). Over the NCP, near-surface $O_3$ exhibits an inverted V-shaped pattern, with maximum $O_3$ concentrations in summer, minimum in winter (Fig. 4). Over the YRD, $O_3$ presents a bridge shape, with relatively higher concentrations in

spring, summer and autumn (Fig. 4). $O_3$ concentrations over the PRD peak in autumn, with a

minimum in summer (Fig. 4). Similarly, $O_3$ over India exhibits a minimum in summer, with

highest concentrations in winter (Fig. 4).

China and India are influenced largely by monsoonal climates (*Wang et al., 2001*), and the

seasonality of $O_3$ in different regions is affected by wind pattern reversals related to the winter

and summer monsoon systems (*Lu et al., 2018b*). Various monsoon indices have been proposed

to describe the major features of the Asian monsoon, based on pressure, temperature, and wind

fields, etc. In the present study, we adopted the dynamical normalized seasonality monsoon

index (DNSMI) developed by *Li and Zeng (2002)* to explore the influence of monsoon intensity

on the seasonal behavior of $O_3$ in the boundary layer in different regions of China and India.

DNSMI is defined as follows:

$$DNSMI = \frac{\|\overline{V_1} - V_i\|}{\overline{V}} - 2 \quad (1)$$

in which $V_1$ and $V_i$ represent the wind vectors in January, and wind vectors in month $i$,

respectively. $\overline{V}$ denotes the mean of wind vectors in January and July. The norm of a given

variable is defined as:

$$\|A\| = (\int \int |A|^2 dS)^{\frac{1}{2}} \quad (2)$$

where S represents the spatial area of each model grid cell. More detailed information on the

definition is presented in *Li and Zeng (2002)*.

This definition of monsoon proposed by *Li and Zeng (2002)* focuses on wind vectors,

representing the intensity of wind direction alternation from winter to summer. In winter,

northwesterly winds are predominant, then higher DNSMI values indicate stronger alternation

of wind directions. For example, DNSMI values are higher than 5 in coastal regions of South

China and most environments in India (Fig. 5c), suggesting that these regions are influenced

largely by the summer monsoon. The spatial distribution of monsoon precipitation in Fig. S7(c)

also indicates that most areas of India and South China are influenced by summer monsoon.

The alternation of wind vectors (Fig. 5) and precipitation (Fig. S7) from winter to summer

results in changes in upwind areas and abundance of $O_3$ precursors, modulating the severity of

$O_3$ pollution. In summer, the southerly winds containing clean maritime air masses, serve to

reduce the intensity of pollution in regions that are affected largely by the summer monsoon

(e.g., most regions over India, and coastal regions of China). Besides, summer monsoon can
bring about cloudy and rainy weather conditions (Fig. S7, removement of ozone precursors),
weaker solar radiation, and lower temperature (Lu et al., 2018b), which are not conducive to
photochemical production of $O_3$ (*Lu et al, 2018b; Tang et al., 2013*). The onset of the summer
monsoon is also associated with strong air convergence and uplift, which is not favorable for
the accumulation of $O_3$ and its precursors (*Lu et al, 2018b*).
North China is less influenced by the summer monsoon as suggested by the insignificant
precipitation in summer (Fig. S7c). East China and South China are more affected as suggested
by DNSMI values higher than 0.5 and more abundant precipitation (Fig. 5c and Fig. S7c).
High temperature and stronger solar radiation in summer favor the photochemical production
of $O_3$. As a result, $O_3$ concentrations in the NCP peak in summer, exhibiting an inverted V-
shaped pattern (Fig. 4a). The YRD region is affected moderately by the summer monsoon, with
DNSMI values greater than 0.6 and mean precipitation greater than 7mm/day (Fig. 5c and S7c).
The upwind sources for the YRD in summer include both polluted (south China) and clean
(ocean) regions. Thus, the inhibition of $O_3$ formation in the YRD due to the summer monsoon
does not lead to the annual minima in summer. Because of the favorable weather conditions
(increasing temperature and solar radiation, and low precipitation) in spring and autumn (Fig.
S7d), the seasonality of $O_3$ in the YRD exhibits a bridge shape, consistent with previous
observations within this region (*Tang et al., 2013*). In addition, southerly winds might bring $O_3$
and its precursors from the YRD region in summer (Fig. 5c), which is further quantified in Sect.
4.3. For India and the PRD region, the alternation of wind fields and precipitation begins as
spring approaches (Fig. 5 and Fig. S7). As a result, $O_3$ concentrations decline in response to
input of cleaner air from the ocean and more precipitation. As summer arrives, the intensity of
the monsoon reaches its maximum (Fig. 5c and S7) and concentrations of $O_3$ in both India and
South China decline to reach their annual minima (Fig. 4c and Fig. 4d). As wind direction
changes over the east coast of China from summer to autumn, $O_3$ peaks in autumn in South
China can be attributed also to the outflow of $O_3$ and its precursors from the NCP and YRD
regions (Fig. 5d). This contribution is discussed further also in Sect. 4.3.

**4.2 $O_3$ sensitivity to emissions from individual source sectors**
$O_3$ in the troposphere is formed through complex nonlinear processes involving emissions of
$NO_x$ and VOCs from various anthropogenic, biogenic, and biomass burning sources. We
illustrate in Fig. 6 the sensitivity of seasonal mean $O_3$ concentrations in both China and India
to individual source sectors, patterns that offer important implications for seasonal $O_3$ control
strategies in some highly polluted regions. The sensitivity is defined as the responses of $O_3$
concentration to the elimination of each source sector ($O_{3_{with\ all\ emissions}} - O_{3_{without\ each\ sector}}$).
For China, summer $O_3$ formation is more sensitive to industrial sources than to other
anthropogenic sources, including power, residential, and transport (Fig. 6c and Table 3).
Emissions from the industrial sector are responsible for an enhancement of $O_3$ concentrations
by more than 8 ppb in the NCP and YRD regions in summer (Fig. 6c and Table 3). Using a
similar approach, *Li et al. (2017)* reported that the contribution to $O_3$ from industrial sources
exceeded 30 $\mu g/m^3$ (~15 ppb) in highly industrialized areas, including Hebei, Shandong,
Zhejiang, etc. during an episode in May. *Li et al. (2016)* concluded that the industrial sector
plays the most important role for $O_3$ formation in Shanghai, accounting for more than 35% of
observed concentrations. Adopting a source-oriented chemical transport model, *Wang et al.,*
*(2019)* demonstrated that the industrial source contributes 36%, 46%, and 29% to non-
background $O_3$ in Beijing, Shanghai and Guangdong, respectively.
In the NCP and YRD regions, $O_3$ formation in winter, spring, and autumn reflects negative
sensitivity to the transport and power sectors (Fig. 6 and Table 3). These two sectors dominate
emissions of $NO_x$ in China (*Li et al., 2017*). Removing these sectors would lead to increases in
$O_3$ in VOC-limited regions of east China in winter, spring and fall (less biogenic emissions of
VOCs in these seasons, *Fu et al., 2012*). The ratio of formaldehyde to reactive nitrogen
($HCHO/NO_y$) is widely used to determine the $O_3$ production sensitivity with critical value of
0.28 (*Sillman, 1995; Zhao et al., 2009*). Fig. S8 indicates that east China is VOCs-limited in
winter, spring and fall. Urban regions in China are still VOC-limited (Fig. S8c, *Fu et al., 2012;*
*Jin et al., 2017*) in summer, leading to negligible or negative sensitivity to the transport and
power sectors as shown in Fig. 6g and Fig 6o. In other regions of east China, removing transport
and power sources would lead to an increase in $O_3$ concentrations by about 4 ppb in summer.
The negative sensitivity of $O_3$ to the transport and power sectors may be also caused by the

nighttime titration effects. In winter, daytime mean $O_3$ exhibit also negative sensitivity to transportation sector, and similar distribution with daily mean $O_3$ sensitivity (Fig. S9a and S9b), suggesting nighttime titration effects might not be the major reason in winter. However, daytime mean and daily mean $O_3$ exhibit different patterns of sensitivity to transportation sector in highly urbanized regions in summer, which could be related to nighttime titration effects. As indicated in Fig. S10, $O_3$ sensitivity to transportation sector in Beijing is positive during the day but negative during the night.

Including biogenic emissions results in an increase in summer mean $O_3$ concentrations by more than 18 ppb in the NCP and YRD regions (Fig. 6s and Table 3). The large sensitivity of $O_3$ to biogenic emissions is associated with the massive VOCs emitted from biosphere (Table S2). The amount of biogenic VOCs is comparable to those emitted from all anthropogenic sectors in China and greater than anthropogenic VOCs in India (Table S2). Using a similar approach, *Li et al. (2018)* found that biogenic emissions contributed 8.2 ppb in urban Xi'an. Other source apportionment studies indicate that the contribution of biogenic emissions to $O_3$ formation is about 20% in China (*Li et al., 2016; Wang et al., 2019*). The enhancements due to biogenic emissions are larger over south China during winter, and the significantly impacted regions extend northwards in spring and autumn (Fig. 6q-6t). Biomass burning emissions lead to relatively lower $O_3$ enhancements over China in winter, but they are responsible for an appreciable contribution to $O_3$ pollution (~4 ppb) in east China in summer (Fig. 6w and Table 3). *Li et al. (2016)* suggested that biomass burning sources contribute about 4% to $O_3$ formation in the YRD region in summer. The enhancement due to biomass burning estimated by *Lu et al. (2019)* using a different model indicates lower values in east China.

For India, $O_3$ formation is most sensitive to the transport vehicle sector (~8 ppb) in all seasons (Table 3), slightly higher than it is to the biogenic source (Fig. 6m-6p and Table 3). Among other sectors, the sensitivity of $O_3$ formation to the residential sector is significant in winter as residential sector emits the largest amount of NMVOCs (*Li et al., 2017*), while the influence of biomass burning emissions is negligible.

To further address the issue of nighttime titration effects, we calculated also the sensitivity of daytime $O_3$ formation in July to sectors, and we found that daytime $O_3$ in the NCP and YRD are also most sensitive to industrial and biogenic emissions (Table 4). Among other

anthropogenic sectors, transportation emissions play important roles in the formation of
daytime $O_3$ in China, followed by power generation emissions (Table 4).Our results highlight
the importance of industrial sources and biogenic emissions in $O_3$ formation in east China,
consistent with the conclusions of *Li et al. (2017)*. The significance of other sectors
demonstrated by *Li et al. (2017)* partly disagrees with the current findings. Conclusions from
*Li et al. (2017)* rely on simulations of a one-week episode in May, while our results provide
more information considering different seasons and different highly polluted regions.

**4.3 $O_3$ contribution from individual source regions**
The sensitivity of $O_3$ pollution to individual source sectors discussed in the previous section
provides a quantitative understanding of the relative importance of individual source sectors.
Additionally, information on the contribution of individual source regions to $O_3$ pollution
should provide useful inputs for $O_3$ control strategies. Because of the large computational costs
of sensitivity simulations, we employed the tagging method to examine contributions to $O_3$
pollution from individual source regions. Fig. 7 presents monthly mean concentrations of $O_3$
averaged over the NCP, YRD, PRD and India, with contributions from individual source
regions.
The NCP region is influenced largely by sources outside China, especially in wintertime, which
might be attributed to less local production and a longer $O_3$ lifetime in winter. In winter, sources
outside China are responsible for more than 75% of $O_3$ formation in the NCP region. However,
this contribution declines to about 50% as summer approaches. Using the tagged tracer method
with a global chemical transport model, *Nagashima et al. (2010)* suggested that sources outside
China contributed about 60% and 40% to surface $O_3$ in North China in spring and summer,
respectively. Our estimate for the contributions of sources outside China in these two seasons
suggests slightly higher values: 73% and 51% (Table 5). In summer, NCP local sources
contribute about 31%, with additional 8% from Northwestern China.
For the YRD region, local emissions contribute 32% to $O_3$ formation in summer, but the
contribution declines by 8% in spring and autumn (Table 5). The contribution of sources
outside China decreases greatly in summer (46%), leading to a small summer $O_3$ trough. The
source apportionment results in *Nagashima et al. (2010)* also indicated that the contribution of
sources outside China to $O_3$ in the Yangtze River Basin decreases significantly from spring to
summer (44% to 30%). The relatively lower contribution from sources outside China is
associated with the prevailing winds and cleaner air from the ocean in summer (Fig. 5c). In
addition to local sources, we further identified the major source region for $O_3$ in the YRD region
is the NCP in winter, spring and autumn (14%, 6% and 8%, respectively). In summer, the major
source region of $O_3$ in the YRD region is Southeast China (10%). *Gao et al. (2016a)* concluded
that YRD local emissions contribute 13.6%-20.6% to daytime $O_3$ under different wind
conditions, and the contribution of super regional sources (Outside) ranges from 32 to 34% in
May. In Hangzhou (a megacity within YRD), source apportionment results reveal that long-
range transport contributes 36.5% to daily maximum $O_3$ with the overall contribution
dominated by local sources (*Li et al., 2016*).
$O_3$ concentrations in the YRD region are influenced largely by the summer monsoon, and the
prevailing winds from the ocean result in a minimum contribution from polluted regions. The
estimated contribution of sources outside China declines to 46% in summer, which agrees well
with the number 47% inferred from *Nagashima et al. (2010)*. *Li et al. (2012)* applied the OSAT
tool in the CAMx model to apportion $O_3$ sources in south China, and they reported that super-
regional sources contributed 55% and 71% to monthly mean $O_3$ in summer and autumn,
respectively. They pointed out also that regional and local sources play more important roles
in $O_3$ pollution episodes (*Li et al., 2002*). The contribution of local source peaks in summer
(41%) exceeds the local contribution in the NCP and YRD regions. As discussed in Sect. 4.1,
the outflow of $O_3$ and its precursors from the NCP and YRD regions might play important roles
in peak autumn $O_3$ in the YRD (Fig. 5d), as wind direction switches from summer to autumn.
We identified the major upwind regions for the PRD in autumn as YRD (11%) and Southeast
China (10%). From summer to autumn, the contribution of YRD sources to the PRD increases
from 2% to 11%. For India, $O_3$ concentrations are dominated by sources outside India, and
sources in North India (Fig. 7d). In winter, sources outside India contribute 49%, while sources
in North India contribute 38%.
We calculated also the contributions of sources in different regions to MDA8 $O_3$ concentrations,
and we compared the results with contributions to daily mean $O_3$. As shown in Fig. 8, the
contributions of sources in different regions do not exhibit a large difference for Beijing, except
that the local sources play a more important role in the formation of daytime $O_3$ in winter (Fig.
8a). Similarly, higher contributions of local sources to the formation of daytime $O_3$ are found
for Guangzhou in autumn, and for Shanghai in all seasons (Fig. 8). The contributions of sources
in different regions do not show a notable difference for New Delhi, India.
The estimated contributions of sources outside China to $O_3$ pollution in receptor regions exhibit
slightly higher values than the values inferred from studies using global models (*Nagashima et*
*al., 2010; Wang et al., 2011*). This might be related partly to the inconsistency between
simulations from the applied regional model and boundary conditions from another global
model. Global chemical transport models usually show better skills in simulating
transboundary pollution.

**5 Discussion and Summary**
In this study, we used a fully coupled regional meteorology-chemistry model with a horizontal
grid spacing of 60 km × 60 km to study the seasonality and characteristics of sources of $O_3$
pollution in highly polluted regions in both China and India. Both observations and model
results indicate that $O_3$ in the NCP, YRD, PRD, and in India display distinctive seasonal
features. Surface concentrations of $O_3$ peak in summer in the NCP, in spring in the YRD, in
autumn in the PRD and in winter in India. These distinct seasonal features for different regions
are linked to the intensity of the summer monsoon, to sources, and to atmospheric transport.
With confidence in the model's ability to reproduce the major features of $O_3$ pollution, we
examined the sensitivity of $O_3$ pollution to individual anthropogenic emission sectors, and to
emissions from biogenic sources and from burning of biomass. We found that production of $O_3$
in summer is more sensitive to industrial and biogenic sources than to other source sectors for
China, while the transportation and biogenic emissions are more important for all seasons in
India. For India, in addition to transportation, the residential sector also plays an important role
in winter when $O_3$ concentrations peak. These differences in conditions between China and
India suggest differences in control strategies on economic sectors should be implemented to
minimize resulting pollution.
Tagged simulations suggest that sources in east China play an important role in the formation
of the summer $O_3$ peak in the NCP, and sources from Northwest China should not be neglected
to control summer $O_3$ in the NCP. For the YRD region, prevailing winds and cleaner air from
the ocean in summer lead to reduced transport from polluted regions, and the major source
region in addition to local sources is Southeast China. For the PRD region, the upwind region
is replaced by contributions from polluted east China as autumn approaches, leading to an
autumn peak. The major upwind regions in autumn for the PRD are YRD (11%) and Southeast
China (10%). For India, sources in North India show larger contributions than sources in South
India.
The focus of our analysis is on the seasonality of $O_3$ pollution and its sources in both China
and India, with an emphasis on implications for $O_3$ control strategies. Most previous studies
focused on the analysis of episodes or monthly means for a region, while the current study
presents a more comprehensive picture. For the NCP region, $O_3$ concentrations peak in summer,
during which industrial sources should be given higher priority. Besides local sources in the
NCP, sources from Northwest China play also important roles. For the YRD region, $O_3$
concentrations in spring, summer and autumn are equally important, showing appreciable
sensitivity to the industrial sources. In addition to local sources, sources from the NCP should
be considered for control of $O_3$ in spring and autumn, while sources from Southeast China
should be considered in summer. For the PRD region, $O_3$ concentrations peak in spring and
autumn, during which reducing industrial and transportation sources could be more effective.
In both spring and autumn, sources from the YRD and Southeast China show appreciable
contributions to $O_3$ pollution in the PRD. For India, $O_3$ pollution is more serious in winter,
during which controlling residential and transport sources in North India could be more
effective.
However, uncertainties remain in the conclusions resulting from the assumptions and
methodology adopted in this study. The zero-out method is computationally inefficient. It is a
sensitivity method, and does not provide source contribution for nonlinear systems, as the sum
of impacts of all sources will not equal the total concentration (*Yarwood et al., 2007*). Although
there is no perfect source apportionment technique for nonlinear systems, reasonable method
that tracks mass contributions and accounts for chemical nonlinearity can provide additional
information in terms of the design of control strategies (*Yarwood et al., 2007*). In the tagging

method, photochemical indicator HCHO/NO$_y$ with threshold of 0.28 (*Sillman, 1995*) was used to determine NO$_x$- or VOC-limited, which can also result in uncertainties in the results. There are several other indicators have been proposed to indicate photochemical sensitivity, including O$_3$/NO$_x$, O$_3$/NO$_y$, etc. However, the robustness of these indicators can vary with ambient conditions and locations (*Andreani-Aksoyoglu et al., 2001*). *Zhang et al. (2009)* recommended using multiple indicators rather than a single one to reduce uncertainties. *Wang et al. (2019)* suggested that the use of a single threshold for these indicators is insufficient, as O$_3$ can be sensitive to both NO$_x$ and VOCs. A three-regime O$_3$ attribution technique was developed by *Wang et al. (2019)* to address this problem. Additionally, although comparisons are shown for daytime mean and daily mean, most conclusions in this study are based on seasonal mean (both daytime and nighttime) O$_3$ while many previous studies investigate sources of 8-h or daily maximum O$_3$. As illustrated in *Li et al. (2016)*, the dominant contribution to nighttime O$_3$ is associated with long-range transport. All of these factors contribute to uncertainties in the results of source apportionment, but should not downplay the significance of current findings in terms of policy implications.

**2 tables and 10 figures are listed in the supplement.**

**Author contribution**

MG and MBM designed the study; MG performed model simulations and analyzed the data with the help from JG, BZ, RK, XL, SS, YZ, BJ, PW, PS; GB, JH, QY, HZ provided measurements. MG and MBM wrote the paper with inputs from all other authors.

**Data availability**

The measurements and model simulations data can be accessed through contacting the corresponding authors.

**Competing interests**

The authors declare that they have no conflict of interests.

**Acknowledgement**

This work is supported by the special fund of Key Laboratory for Aerosol-Cloud-Precipitation of China Meteorological Administration, Nanjing University of Information Science and Technology (KDW1901), Harvard Global Institute, special fund of State Key Joint Laboratory of Environment Simulation and Pollution Control (19K03ESPCT), the Natural Science Foundation of Guangdong Province (no. 2019A1515011633), National Natural Science Foundation of China Major Research Plan (Integrated Project) (NSFC91843301), and the National Key Research and Development Program-Cooperation on Scientific and Technological Innovation in Hong Kong, Macau and Taiwan (2017YFE0191000).

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

Table 1. Descriptions of simulations

| Simulations | Descriptions |
|---|---|
| Control | Anthropogenic, biogenic and fire emissions are considered; |
| Industrial | Same as control except industry sector in anthropogenic emissions is excluded; |
| Residential | Same as control except residential sector in anthropogenic emissions is excluded; |
| Transportation | Same as control except transportation sector in anthropogenic emissions is excluded; |
| Power | Same as control except power sector in anthropogenic emissions is excluded; |
| Biogenic | Same as control except biogenic emissions are excluded; |
| Fire | Same as control except fire emissions are excluded; |

Table 2 Model evaluation statistics

| Regions | NCP | YRD | PRD | India |
|---|---|---|---|---|
| Mean Bias | -3.8 | -1.8 | 3.1 | -2.0 |
| Root Mean Square Error | 6.4 | 5.5 | 7.9 | 4.4 |
| Normalized Mean Bias | -13.3% | -6.2% | 10.7% | -5.6% |
| Normalized Mean Error | 18.7% | 14.9% | 21.2% | 11.1% |
| R | 0.98 | 0.96 | 0.84 | 0.91 |

Table 3. Sensitivity of seasonal $O_3$ to emission sectors for different regions (ppb)

| Sectors | Seasons | NCP | YRD | PRD | India |
|---------|---------|------|------|------|-------|
| Industry | Winter | -4.1 | -1.5 | 4.5 | 2.1 |
| | Spring | -0.3 | 3.8 | 6.5 | 1.7 |
| | Summer | 8.3 | 8.3 | 4.7 | 1.6 |
| | Autumn | -1.4 | 1.7 | 7.1 | 2.1 |
| Power | Winter | -5.6 | -7.5 | -1.2 | 1.7 |
| | Spring | -3.2 | -2.2 | 2.2 | 2.3 |
| | Summer | 2.7 | 2.9 | 3.3 | 1.9 |
| | Autumn | -3.3 | -3.3 | 2.1 | 2.4 |
| Residential | Winter | 5.1 | 7.7 | 6.6 | 4.2 |
| | Spring | 2.4 | 2.5 | 1.9 | 2.4 |
| | Summer | 2.5 | 1.4 | 1.1 | 2.2 |
| | Autumn | 2.2 | 2.2 | 1.6 | 3.2 |
| Transport | Winter | -8.5 | -8.0 | 0.2 | 7.6 |
| | Spring | -3.7 | -1.5 | 3.4 | 7.9 |
| | Summer | 2.8 | 4.0 | 3.6 | 6.7 |
| | Autumn | -4.3 | -3.3 | 3.0 | 8.9 |
| Biogenic | Winter | 0.3 | 1.0 | 3.8 | 4.8 |
| | Spring | 4.3 | 6.6 | 7.5 | 5.6 |
| | Summer | 19.2 | 18.5 | 9.4 | 5.7 |
| | Autumn | 5.7 | 6.5 | 11.4 | 8.0 |
| Fire | Winter | 0.1 | 0.2 | 2.3 | 0.6 |
| | Spring | 1.1 | 1.8 | 2.6 | 1.1 |
| | Summer | 3.8 | 4.0 | 1.2 | 0.2 |
| | Autumn | 1.2 | 1.4 | 1.9 | 0.5 |




Table 4. Sensitivity of summer (July) daytime O₃ to emission sectors for different regions (ppb)

| Sectors | NCP | YRD | PRD | India |
|---|---|---|---|---|
| Industry | 19.9 | 14.3 | 7.1 | 2.3 |
| Power | 6.1 | 7.0 | 4.9 | 2.7 |
| Residential | 4.1 | 1.9 | 1.6 | 3.3 |
| Transport | 8.9 | 9.2 | 5.9 | 10.0 |
| Biogenic | 28.7 | 28.9 | 12.0 | 7.6 |
| Fire | 1.4 | 0.8 | 0.3 | 0.1 |

Table 5. Long range transport, local, and regional source contributions for seasonal mean O3 for different regions

| | NCP | YRD | PRD | India |
|---|---|---|---|---|
| Winter | Outside: 81% | Outside:51% | Outside: 44% | Outside: 49% |
| | Local: 12% | Local: 26% | Local: 13% | N India: 35% |
| | NW China: 6% | NCP: 14% | YRD: 13% | S India: 16% |
| Spring | Outside: 73% | Outside:59% | Outside: 48% | Outside: 58% |
| | Local: 17% | Local: 24% | Local: 27% | N India: 28% |
| | NW China: 5% | NCP: 6% | YRD: 7% | S India: 14% |
| | | | SE China: 6% | |
| Summer | Outside: 51% | Outside:46% | Outside: 46% | Outside: 45% |
| | Local: 31% | Local: 32% | Local: 41% | N India: 38% |
| | NW China: 8% | SE China: 10% | SE China: 4% | S India: 17% |
| Autumn | Outside: 69% | Outside:61% | Outside: 50% | Outside: 42% |
| | Local: 21% | Local: 24% | Local: 15% | N India: 41% |
| | NW China: 7% | NCP: 8% | YRD: 11% | S India: 17% |
| | | | SE China: 10% | |

(Outside sources represent sources outside China for the discussed three regions in China, and sources outside

India for India, including also transport from upper boundary of the model; NCP: Beijing, Tianjin, Hebei,
Shandong, and Henan; YRD: Anhui, Jiangsu, Shanghai and Zhejiang; SE China: Jiangxi, Fujian and Taiwan;
Central China: Hunan and Hubei; South China: Guangxi and Hainan)


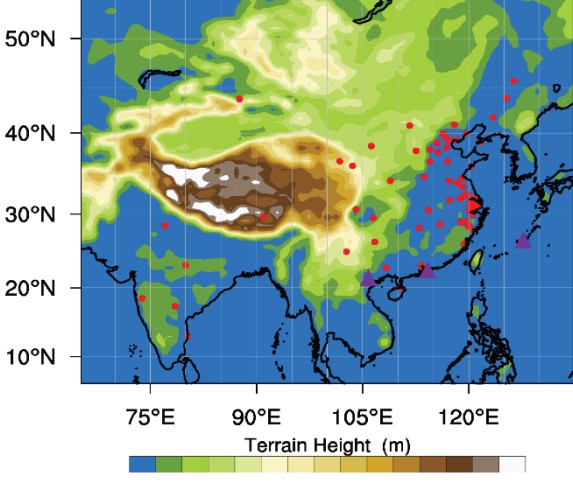


Fig. 1. WRF-Chem domain setting with terrain height and the locations of surface ozone
observations marked by solid red circles. Purple solid triangles mark the location of
ozonesonde observations.

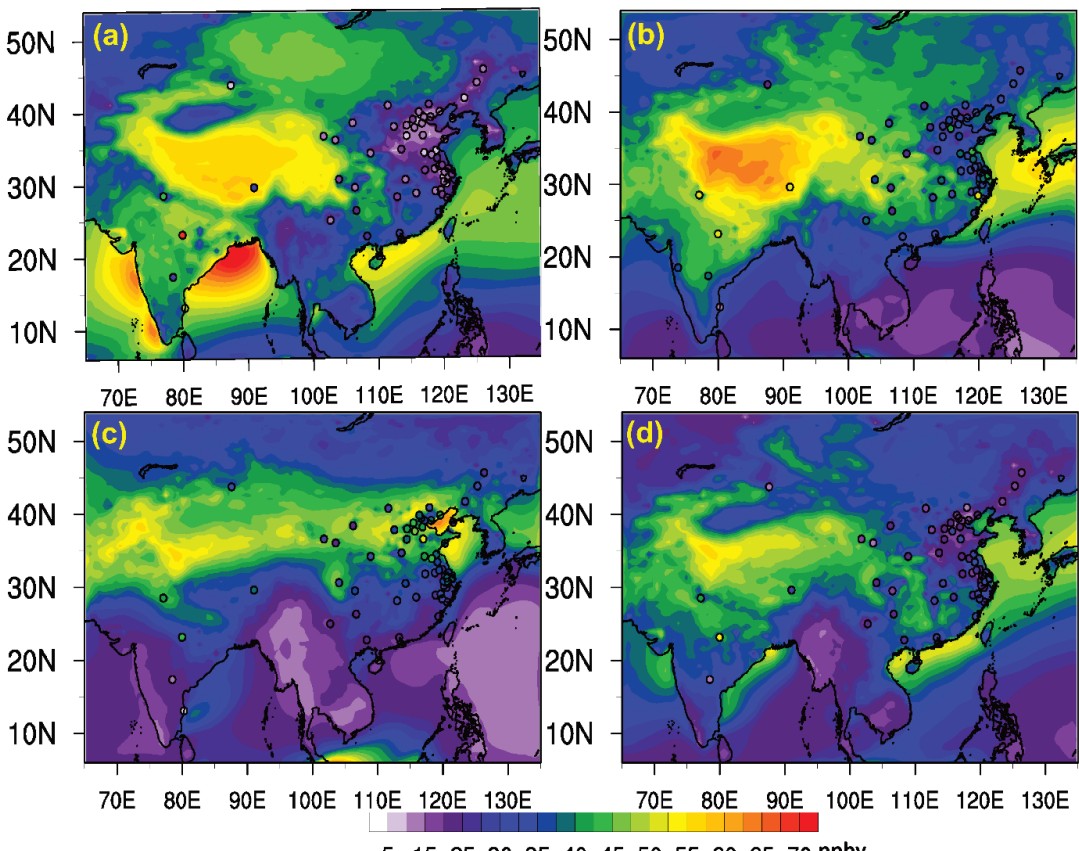


Fig. 2. Spatial distribution of simulated and observed seasonal mean ozone concentrations for
Winter (a), Spring (b), Summer (c) and Fall (d).

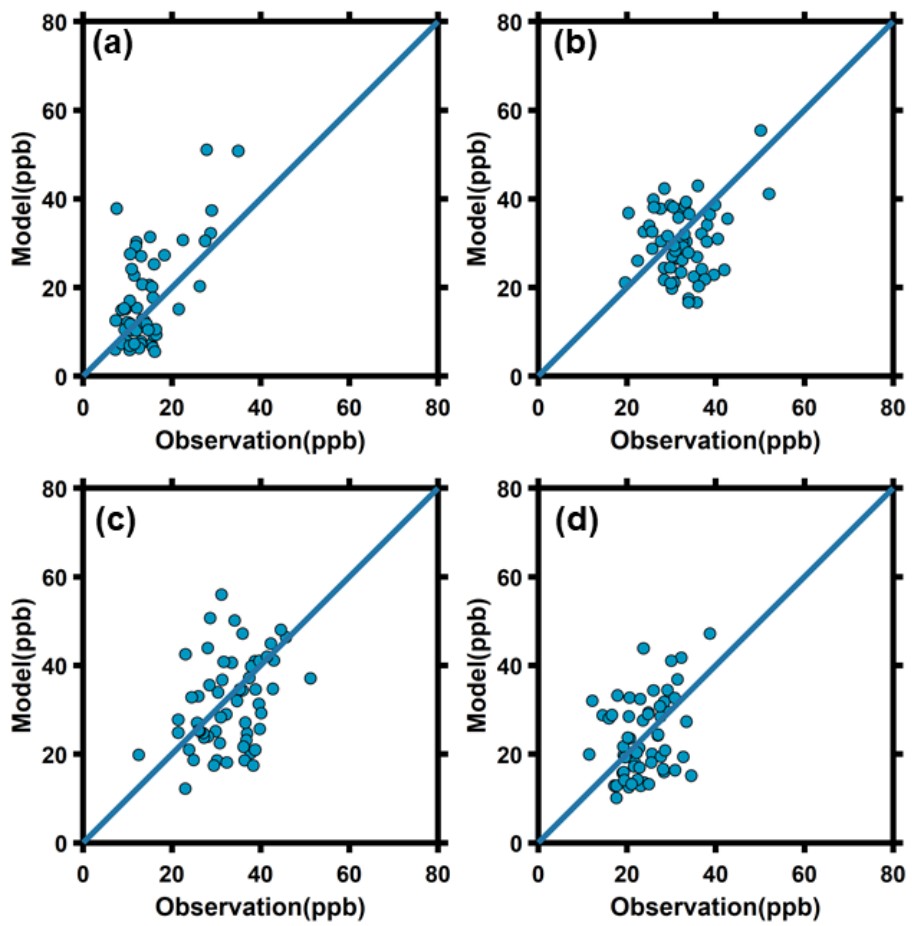


Fig. 3. Scatter plot of simulated and observed seasonal mean ozone concentrations for Winter
857                    (a), Spring (b), Summer (c) and Fall (d).


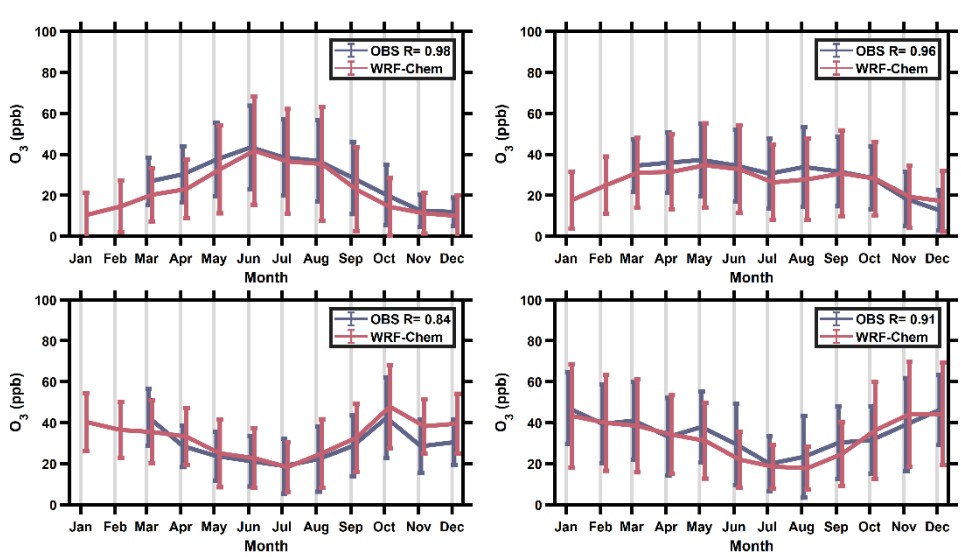


Fig. 4. Observed and simulated monthly mean $O_3$ concentrations averaged for the North
China Plain (NCP) (a), Yangtze River Delta (YRD) (b), Pearl River Delta (PRD) (c), and
862                                India (d).




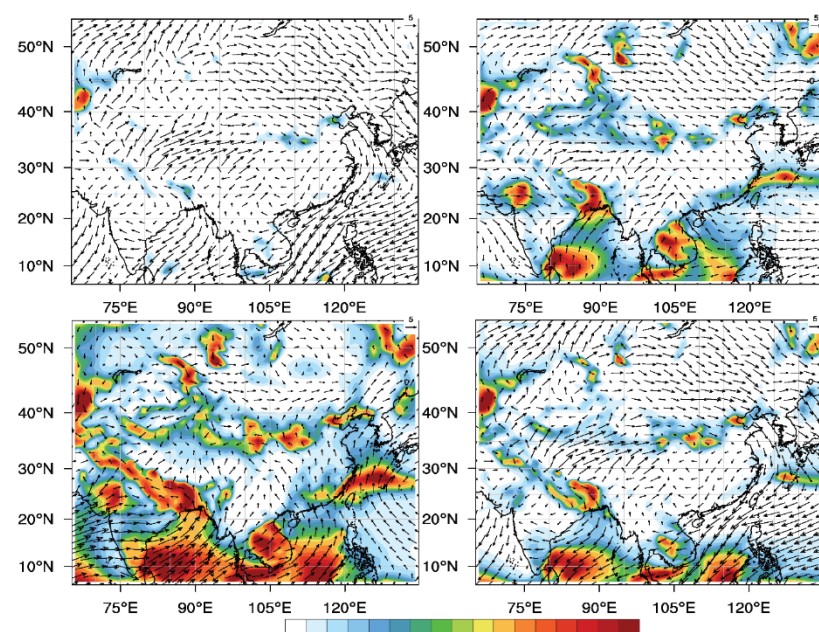


Fig. 5. Modeled mean near surface wind fields (winds at 10 meters above ground) and the
monsoon index in the boundary layer (0-1.5km) for winter (December, January, and February,

869        a), spring (March, April, and May, b), summer (June, July and August, c), and autumn

870                  (September, October, and November, d).




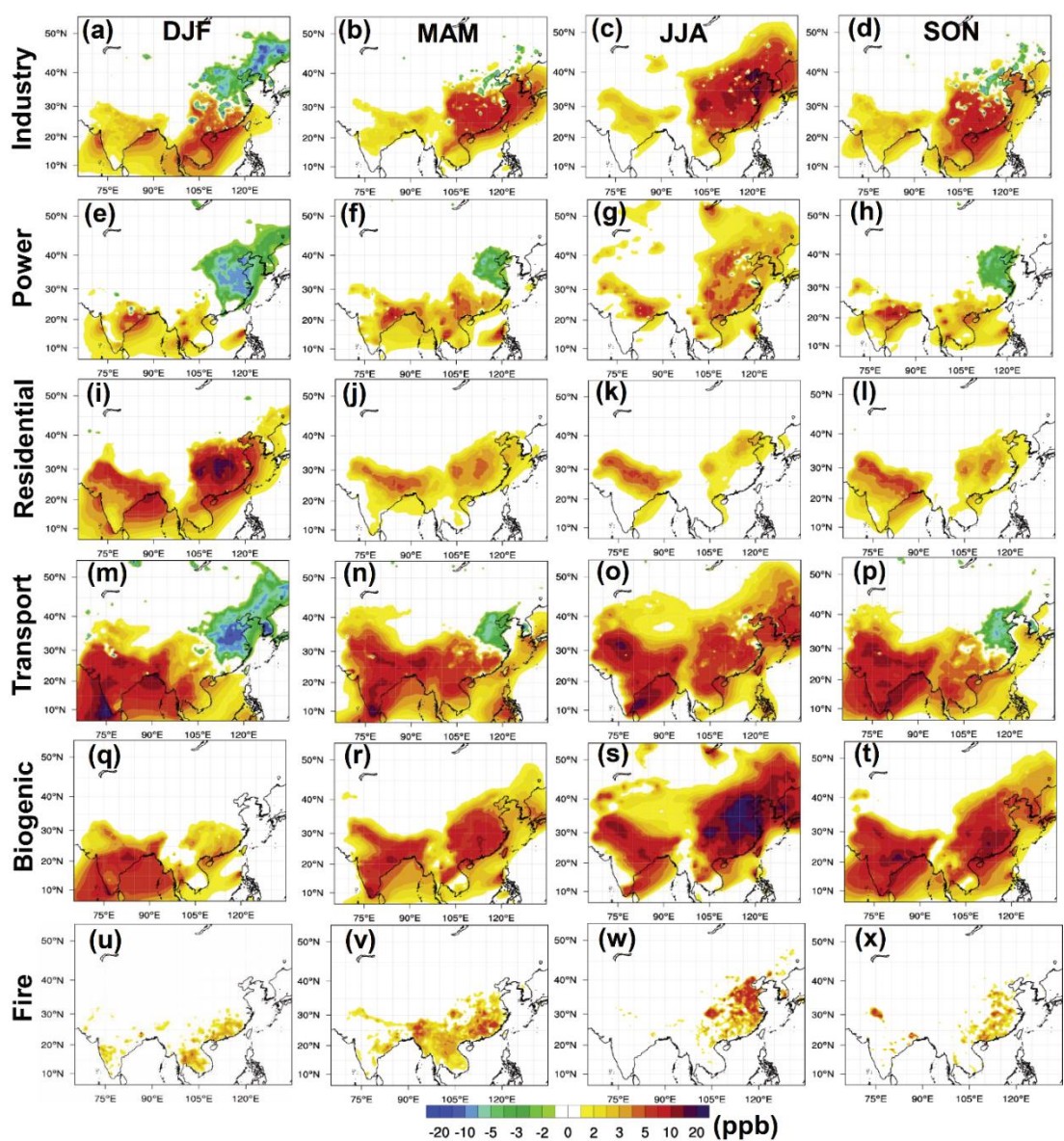


Fig. 6. Distributions of the contributions to near-surface ozone averaged for winter, spring,
summer and autumn from industry (a-d), power, (e-h), residential (i-l), transport (m-p),
biogenic (q-t) and fire/biomass burning (u-x) emissions.

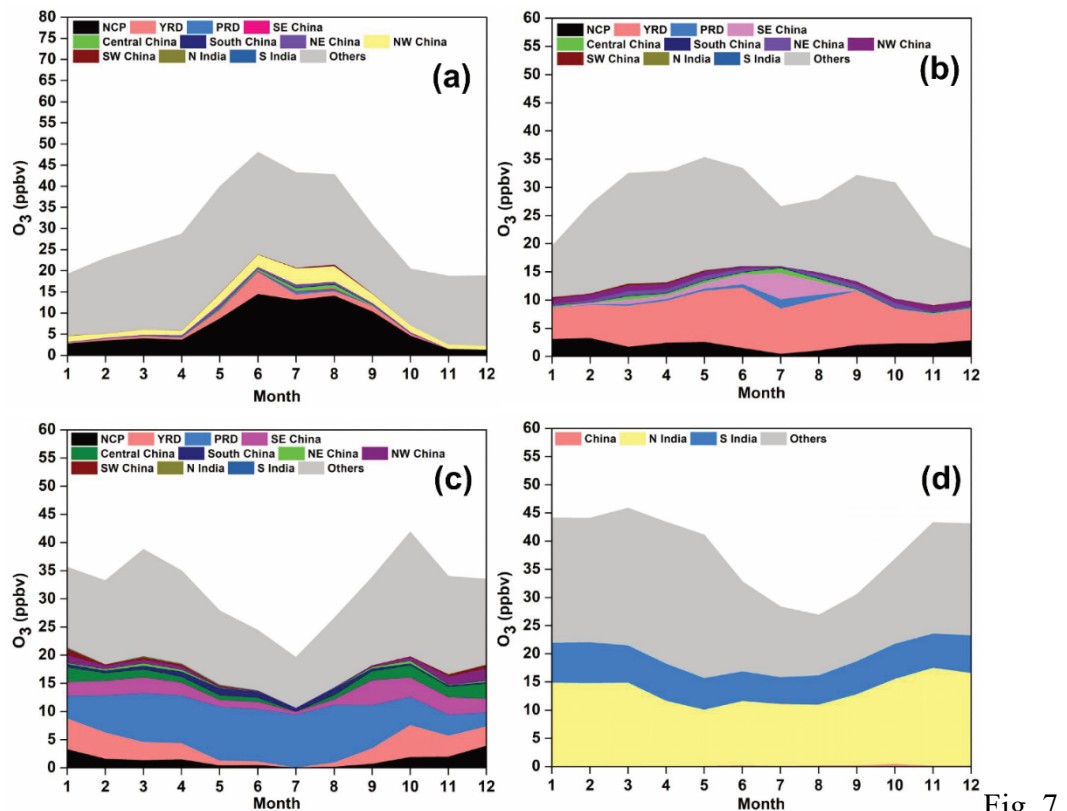

Fig. 7.
Contributions to monthly mean ozone in NCP (a), YRD (b), PRD (c), and India (d) from
different source regions (NCP: Beijing, Tianjin, Hebei, Shandong, and Henan; YRD: Anhui,
Jiangsu, Shanghai and Zhejiang; SE China: Jiangxi, Fujian and Taiwan; Central China:
Hunan and Hubei; South China: Guangxi and Hainan).

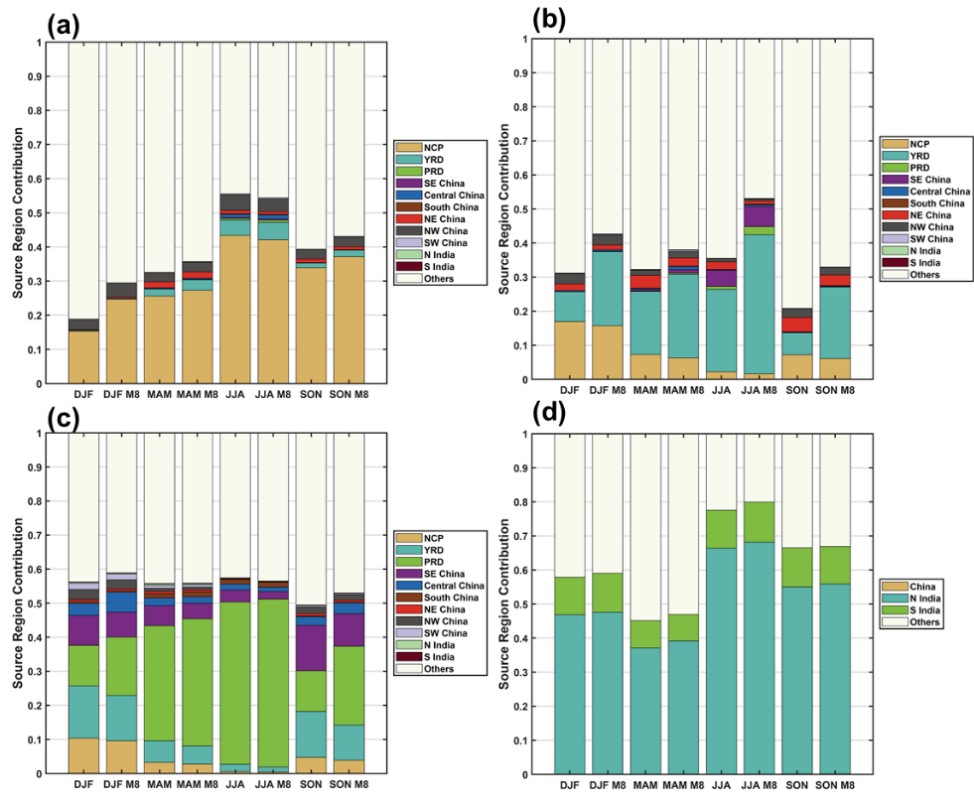

Fig. 8. Contributions to seasonally daily mean ozone (DJF, MAM, JJA, and SON) and MDA8 ozone (DJF M8, MAM M8, JJA M8, and SON M8) in Beijing (a), Shanghai (b), Guangzhou (c), and New Delhi (d) from different source regions.