# Peer review of "Ozone Pollution over China and India: Seasonality and Sources"

_Atmospheric Chemistry and Physics, 2019_

## Referee Comment (RC1) · Anonymous Referee #1 · 6 Dec 2019

Review comments on "Ozone Pollution over China and India: Seasonality and Sources" by Gao et al.

This manuscript provided a study investigating the seasonality of ozone pollution and its sources in China and India. This study found different seasonal futures in NCP, YRD, PRD regions of China, South and North of India. This manuscript pointed out that different emission sectors dominate the ozone formation in individual region. The topic is applicable for Atmospheric Chemistry and Physics. The text is concisely written and well documented. However, this manuscript lacked lots of details which are important for the conclusions. First, the authors discussed that the WRF-Chem model may not correctly represent the nighttime ozone chemistry, which leads to the high bias of nighttime ozone levels. If that is the case, in my opinion the study should focus on the daytime ozone concentrations (such as daily maximum 8-hr ozone) when the ozone formation is sensitive to the emissions. The daily mean ozone concentrations used in this study could not correctly reflect the photochemical formation of tropospheric ozone, but might include the effects from nighttime ozone destruction. For instance, the negative contribution from Power and Transportation emissions may be due to the nighttime titration of NO-O3, but may not come from the non-linear chemistry of ozone production as suggested in the manuscript. Second, this manuscript extensively discussed the sensitivity of seasonal ozone to individual source (Figure 5), the authors calculated the contribution from the sensitivity experiments (Table 1) using the 'zero-out' method. Due to the highly non-linear characteristics of the ozone production, it may introduce some uncertainty. The manuscript stated that Sillman (1995) value was used to identify the NOx or VOC-limited region, but I did not notice where the ozone chemical regime was discussed. The value from Sillman (1995) study on the U.S. may not be applicable for China. Third, the current Section 5 lacked discussion of the uncertainty of this study and potential weakness of the modeling platform. Revision of the Summary is suggested. Lastly, there are some missing and errors in the references, which need to be corrected in the revised manuscript. In summary, major revisions as indicated in the comments and remarks below are needed before consideration of publication in ACP.

Detailed Remarks/Suggestions for Revision

Line 98: 'Lu et al. 2018a' is not listed in the bibliography. Two 'Lu et al. 2018' are listed in the reference, so the revised manuscript should list 'a' and 'b' in the bibliography.

Line 94: Stratospheric intrusions can only influence surface ozone levels in high altitude region, while most the regions in both China and India discussed in this manuscript may not be able to see this impact.

Line 97 'J. Gao et al., 2016' is not the correct format for ACP. Please correct all of them.

Line 97: 'Lu et al. 2019' is missing.

Line 122-124: In my opinion, both the spatial and vertical resolutions of the WRF-Chem simulations are too coarse. How many vertical layers of the total 27 layers are in the PBL?

Line 124: A table showing the major physical options is suggested, either in the main article or in the supplementary material.

Line 125: 'M. Gao et al., 2016', same format problem as indicated above.

Line 169: 'Hunan', 'Hubei', and 'Jiangxi' are all labeled 'Hainan' in Figure S1

Line 181: The emissions inventory was developed for year 2012, so why the year of 2013 was simulated?

Line 207: As mentioned above, the authors should try variables such as daily maximum 8-hr ozone for the evaluation.

Line 211: It is hard to figure out if the model can capture the site observations. A scatter plot is needed here.

Line 215: Figure S2 needs more explanation. How the OMI NO2 product was used here? Criteria to filter the row anomaly? What is the solar zenith angle used to filter out data here? When computing the WRF-Chem NO2 columns, how the model results are sampled? For instance, all the grids collocated with missing OMI data points due to cloud in OMI NO2 should not be used. How the OMI averaging kernel was used here? A detailed explanation is suggested.

Line 217: In Figure S2, it is hard to conclude that the NO2 column from WRF-Chem is less than OMI NO2 column. A different plot or ratio plot is needed here. Also the emission deficiency may not be able to fully explain the high bias of nighttime ozone, because the nighttime nitrogen chemistry may not be explicit in WRF-Chem. Again, I suggest to re-do the analysis using daytime ozone to eliminate the impacts from nighttime ozone.

Line 261: 'Lu et al., 2018b'?

Line 287: This figure shows the Beijing-Tianjin-Hebei area is significantly influenced.

Line 292-293: The change of circulation is important because of the regional transport of air pollutants, but the change of cloudiness and precipitation pattern also plays an important role in the formation of ozone.

Line 311: As discussed above, a paragraph is needed here to explain how the sensitivity was calculated.

Line 314: I disagree with this statement. From Fig 5, looks like emissions from Transportation sector contribute more than the Industry sector in China.

Line 317: Li et al. (2017) is missing.

Line 341: Is 'biomass burning' here equivalent to 'Fire' sector in Figure 5?

Line 347: Looks like the biogenic emissions dominate the ozone production in India.

Line 352: I disagree with this statement. Due to the VOC-limited ozone production, biogenic emissions has the largest contribution (Figure 5S).

Line 771 Table 2: Does the 'Outside' stand for outside of China? Need clarification.

Line 779 Figure 1: It is very hard to tell the Purple sold circle in the map. I suggest using a different shape such as solid star or triangle here.

Line 783 Figure 2: As mentioned above, it is hard to see the model performance. A scatter plot is suggested here.

Line 787 Figure 3: The scale in y-axis is improper. Please re-plot the figure with y-axis from 0 to 80 ppbv or 100 ppbv. Also the current plot shows little difference between these two lines. A table shows some statistics such as NMB and RMSE is suggested.

Line 797 Figure 4: what is the 'near surface wind field' here? 10 m or 850 hPa wind?

---

## Referee Comment (RC2) · Anonymous Referee #2 · 9 Dec 2019

**Review of the manuscript titled "Ozone Pollution over China and India: Seasonality and Sources."** (By Meng Gao et al.)

The paper presents the results of an WRF-Chem modeling and analyzed the seasonality of O3 and its sources in both China and India. They derived the modeling results from the sensitivity tests (authors named this as 'a factor separation approach' ) and they explained that the importance of industrial sources in summer in China and the transport vehicle sector in all seasons in India. Also authors described the contributions from other regions.

The content of the manuscript is reasonable and the results also seem convincing. However, all the discussion is mostly based on modeling results, and thus the evaluation and validation of WRF-Chem results have to be made more rigorously. Also some of the descriptions in the text should be changed to be in more quantitative way, and authors can provide a statistical and quantitative modeling (or observational) results on the relations between O3 and its precursors.

 (Major Comments)

1) Line 166, 23 source regions: In most figures, regions are divided into four areas (NCP, YRD, PRD, and India) for current analysis. However, in Fig. S1, authors provided only province borders in China. Please redraw the boundaries of four regions (and 8 additional regions, as employed in Fig. 6) in the Figures.

2) In Fig. 2, in winter, for example, it is considered that the O3 concentration level in East China shows considerable biases. It is suggested that at least the statistical results of four regions (NCP, YRD, PRD, and India) should be explored. Statistical results such as RMSE, IOA, etc (together with correlation coefficients as indicated in Line 228) should be tabulated (or described in the manuscript) for at least this four regions.

3) In Fig. 3, if the y-axis scale is lowered to 80 ppb (or even lower), the differences (between modeling and observations) is unlikely to be ignored. It appears that the spring season (Mar. Apr, May) in NCP is so low that the model result looks nearly doubled, and also 3 months (Oct Noc Dec) in PRD show considerable bias. India may drive higher O3 measurement in April, but the model is not simulated to be as high as measurement. Explain the reasons why the monthly measured average is high in India but why not simulated in April, through WRF-Chem model.

4) In this study, as a Monsoon Index, only wind field index was chosen (Li and Zeng, 2002). Of course, we know that the wind field is an important factor that causes precipitation by the Monsoon cycle, but the distribution of precipitation itself is one of the most important controlling factors in dealing with O3. Collect reanalyzed precipitation data (i.e., GPCP data) and plot horizontal distributions in supplementary material, and analyze both precipitations vs. O3 modeling patterns to explore the regional characteristics more directly.

5) It seems reasonable to analyze O3 during the night and day separately, to see the over- (or under-) estimation of titration effects, day and night separately, as referee 1 pointed out.

6) Add quantitative emissions for gas such as SO2 NOx, NH3, and VOCs together with particulate matters

for the whole 2013 year. Also, describe quantitatively the total of biogenic emission and biomass burning emission over both China and India for 2013 year, and compare to other emissions (i.e., SO2 NOx, NH3, and VOCs)

 (Minor Comments)

1) Line 133: Year 2010 isn't so old ? The changes in emissions should be mentioned briefly in the revised manuscript.

2) Line 245 Fig. S3 → Fig. S4?

3) Line 246, satellite NO2 column (Fig. S2) indicates only day time ? If this is the case, in Fig. S2, modeling results also should be only day-time results?

4) Do not you have any VOC observations?

---

## Author Response (AR1)

**Referee #1**

Review comments on "Ozone Pollution over China and India: Seasonality and Sources" by Gao et al.

This manuscript provided a study investigating the seasonality of ozone pollution and its sources in China and India. This study found different seasonal futures in NCP, YRD, PRD regions of China, South and North of India. This manuscript pointed out that different emission sectors dominate the ozone formation in individual region. The topic is applicable for Atmospheric Chemistry and Physics. The text is concisely written and well documented. However, this manuscript lacked lots of details which are important for the conclusions.

First, the authors discussed that the WRF-Chem model may not correctly represent the nighttime ozone chemistry, which leads to the high bias of nighttime ozone levels. If that is the case, in my opinion the study should focus on the daytime ozone concentrations (such as daily maximum 8-hr ozone) when the ozone formation is sensitive to the emissions. The daily mean ozone concentrations used in this study could not correctly reflect the photochemical formation of tropospheric ozone, but might include the effects from nighttime ozone destruction. For instance, the negative contribution from Power and Transportation emissions may be due to the nighttime titration of NO-O3, but may not come from the non-linear chemistry of ozone production as suggested in the manuscript.

Second, this manuscript extensively discussed the sensitivity of seasonal ozone to individual source (Figure 5), the authors calculated the contribution from the sensitivity experiments (Table 1) using the 'zero-out' method. Due to the highly non-linear characteristics of the ozone production, it may introduce some uncertainty. The manuscript stated that Sillman (1995) value was used to identify the NOx or VOC-limited region, but I did not notice where the ozone chemical regime was discussed. The value from Sillman (1995) study on the U.S. may not be applicable for China.

Third, the current Section 5 lacked discussion of the uncertainty of this study and potential weakness of the modeling platform. Revision of the Summary is suggested. Lastly, there are some missing and errors in the references, which need to be corrected in the revised manuscript. In summary, major revisions as indicated in the comments and remarks below are needed before consideration of publication in ACP.

> ➢ Reply: We would like to thank the review for careful reading and valuable comments. Following your suggestions, we have added more details in the revised manuscript.
> ➢ We added model evaluation of diurnal pattern of ozone in three regions in China to check if the nighttime titration is underestimated or overestimated by the model. We also added model evaluation of MDA8 in Beijing and results based on daytime $O_3$. 24-h mean concentrations are frequently used for short-term health exposure assessment (Lefohn et al., 2018) and they are also important model-measurement comparison metrics (Lefohn et al., 2018). However, we agree that it cannot

represent ozone chemical formation and may confuse the readers on the roles of power and transportation emissions. To address this issue, we conducted additional simulations to examine the diurnal variations of the contribution from transportation emissions in winter and summer. We also added discussion on comparisons between region contrition to 24-hr mean and MDA8 of ozone. (**ref. Fig. S3, Fig. S4, Fig. S9, Fig. S10, Table 4 and Fig. 8**).

➢ Yes, the zero-out method and Sillman (1995) method can bring about uncertainties, and we added discussion of the uncertainties resulted from zero-out method and the method of identify sensitivity regime in Section 5: "In the tagging method, photochemical indicator HCHO/NOy with threshold of 0.28 (Sillman, 1995) was used to determine NOx- or VOC-limited, which can also result in uncertainties in the results. There are several other indicators have been proposed to indicate photochemical sensitivity, including O3/NOx, O3/NOy, etc. However, the robustness of these indicators can vary with ambient conditions and locations (Andreani-Aksoyoglu et al., 2001). Zhang et al. (2009) recommended using multiple indicators rather than a single one to reduce uncertainties. Wang et al. (2019) suggested that the use of a single threshold for these indicators is insufficient, as O3 can be sensitive to both NOx and VOCs. A three-regime O3 attribution technique was developed by Wang et al. (2019) to address this problem."

➢ We also added plots to discuss ozone chemical regime. (**ref. Fig. S8**)

➢ Following the suggestions, we corrected the errors in the references as well.

Lefohn, A.S., Malley, C.S., Smith, L., Wells, B., Hazucha, M., Simon, H., Naik, V., Mills, G., Schultz, M.G., Paoletti, E. and De Marco, A., 2018. Tropospheric ozone assessment report: Global ozone metrics for climate change, human health, and crop/ecosystem research. Elementa (Washington, DC), 1, p.1.

Detailed Remarks/Suggestions for Revision

Line 98: 'Lu et al. 2018a' is not listed in the bibliography. Two 'Lu et al. 2018' are listed in the reference, so the revised manuscript should list 'a' and 'b' in the bibliography.

➢ Reply: We have added.

➢ Lu, X., Hong, J., Zhang, L., Cooper, O.R., Schultz, M.G., Xu, X., Wang, T., Gao, M., Zhao, Y. and Zhang, Y.: Severe surface ozone pollution in China: A global perspective, Environ. Sci. Technol. Lett., 5, 8, 487-494, https://doi.org/10.1021/acs.estlett.8b00366, 2018a.

➢ Lu, X., Zhang, L., Liu, X., Gao, M., Zhao, Y., and Shao, J.: Lower tropospheric ozone over India and its linkage to the South Asian monsoon, Atmos. Chem. Phys., 18, 3101–3118, https://doi.org/10.5194/acp-18-3101-2018, 2018b.

Line 94: Stratospheric intrusions can only influence surface ozone levels in high altitude region, while most the regions in both China and India discussed in this manuscript may not be able to see this impact.

➢ Reply: Yes. We agree that stratospheric intrusions can only influence surface ozone levels in high altitude region, such as Tibet Plateau in China, and populated regions are not largely influenced. Kumar et al. (2010) reported results based on a site in high altitude region, which might not be suitable for discussion of ozone in highly polluted regions here.
➢ To avoid confusion, we removed stratospheric intrusion in the revised manuscript.

Line 97 'J. Gao et al., 2016' is not the correct format for ACP. Please correct all of them.

➢ Reply: We have updated with correct format.

Line 97: 'Lu et al. 2019' is missing.

➢ Reply: We have added it in the revised manuscript.
➢ Lu, X., Zhang, L., Chen, Y., Zhou, M., Zheng, B., Li, K., Liu, Y., Lin, J., Fu, T.-M., and Zhang, Q.: Exploring 2016–2017 surface ozone pollution over China: source contributions and meteorological influences, Atmos. Chem. Phys., 19, 8339–8361, https://doi.org/10.5194/acp-19-8339-2019, 2019.

Line 122-124: In my opinion, both the spatial and vertical resolutions of the WRF-Chem simulations are too coarse. How many vertical layers of the total 27 layers are in the PBL?

➢ Reply: About 10 layers are in the PBL in summer.
➢ In previous examination, Yu et al. (2014) found using 27 vertical layers and 54 vertical layers do not significantly affect simulated concentrations of ozone (Fig. 6.8 in Yu et al., 2014).
➢ In this study, we conducted a large number of sensitivity simulations. To reduce computational complexity and save storage space of the results, we used 27 layers.
➢ Yu, M., 2014. An assessment of urbanization impact in China by using WRF-Chem and configuration optimization.

Line 124: A table showing the major physical options is suggested, either in the main article or in the supplementary material.

➢ Reply: We have added in the supplementary material:

Table S1 Selected Physics Configuration Options

| Feature | Option | Description |
|---|---|---|
| Microphysics | Lin scheme | Lin et al., 1983 |
| Long-wave Radiation | Rapid Radiative Transfer Model (RRTM) | Mlawer et al., 1997 |
| Shortwave Radiation | Goddard shortwave | Chou et al., 1998 |
| Surface Model | Noah Land | Ek et al., 2003 |
| Planetary boundary layer parameterization | Yonsei University | Hong et al., 2006 |

Chou, M.-D., Suarez, M. J., Ho, C.-H., Yan, M. M.-H. and Lee, K.-T.: Parameterizations for cloud overlapping and shortwave single-scattering properties for use in general circulation and cloud ensemble models, J. Climate, 11, 202–214, 1998.

Ek, M.B., Mitchell, K.E., Lin, Y., Rogers, E., Grunmann, P., Koren, V., Gayno, G. and Tarpley, J.D.: Implementation of Noah land surface model advances in the National Centers for Environmental Prediction operational mesoscale Eta model. J. Geophys. Res., 108(D22), 2003.

Hong, S.-Y., Noh, Y., and Dudhia, J.: A New Vertical Diffusion Package with an Explicit Treatment of Entrainment Processes, Mon. Weather Rev., 134, 2318–2341, 2006.

Lin, Y.-L., Farley, R. D., and Orville, H. D.: Bulk parameterization of the snow field in a cloud model, J. Clim. Appl. Meteorol., 22, 1065–1092, 1983.

Mlawer, E. J., Taubman, S. J., Brown, P. D., Iacono, M. J., and Clough, S. A.: Radiative transfer for inhomogeneous atmospheres: RRTM, a validated correlated-k model for the longwave, J. Geophys. Res., 102, 16663–16682, doi:10.1029/97JD00237, 1997.

Line 125: 'M. Gao et al., 2016', same format problem as indicated above.

➢ Reply: We have updated.

Line 169: 'Hunan', 'Hubei', and 'Jiangxi' are all labeled 'Hainan' in Figure S1

➢ Reply: We have updated.

[Figure]

Line 181: The emissions inventory was developed for year 2012, so why the year of 2013 was simulated?

> ➢ Reply: We simulated the year of 2013 because a large number of observations are only available since 2013.
> ➢ When we conducted the simulations, emissions inventory developed for year 2013 were not available.
> ➢ According to the statistics in Zheng et al. (2018), emissions of NMVOCs in 2012 and 2013 were the same, and emissions of NOx only declined by 5%. Thus, we expect that 1 year lag in the inventory will not largely affect our results.

Line 207: As mentioned above, the authors should try variables such as daily maximum 8-hr ozone for the evaluation.

> ➢ Reply: The observation data used in study were taken from Hu et al. (2016). We did not have original hourly data for all stations to calculate MDA8 (maximum daily 8-hr average).
> ➢ We agree MDA8 is quite important for ozone, and we use additional hourly observations of ozone in Beijing to validate MDA8 and add new analysis based on model MDA8 in the revised manuscript. As shown in Figure. S2, the model MDA8 is generally consistent with the observation, except that observed MDA8 ozone concentrations are underestimated in spring.
> ➢ We also added comparison between observed (blue) and modeled (red) monthly averaged diurnal variations of O3 concentrations in the NCP (a), YRD (b) and PRD (c). In the NCP, model overestimates the titration effects, which can partly explain the underestimation of O3 in spring in the NCP (Fig. S3). The titration effects are also overestimated in the YRD, but underestimated in the PRD in winter (Fig. S3).
> ➢ For the results of O$_3$ sensitivity to emission sectors, we added results based on daytime O$_3$ in Table 4: "To further address the issue of nighttime titration effects, we calculated also the sensitivity of daytime O3 formation in July to sectors, and we found that daytime O3 in the NCP and YRD are also most sensitive to industrial and biogenic emissions (Table 4). Among other anthropogenic sectors, transportation emissions play important roles in the formation of daytime O3 in China, followed by power generation emissions (Table 4)."
> ➢ For the tagged ozone results, we added comparison with MDA8 ozone in Fig. 8: "We calculated also the contributions of sources in different regions to MDA8 O3 concentrations, and we compared the results with contributions to daily mean O3. As shown in Fig. 8, the contributions of sources in different regions do not exhibit a large difference for Beijing, except that the local sources play a more important role in the formation of daytime O3 in winter (Fig. 8a). Similarly, higher contributions of local sources to the formation of daytime O3 are found for

Guangzhou in autumn, and for Shanghai in all seasons (Fig. 8). The contributions of sources in different regions do not show a notable difference for New Delhi, India."

[Figure]

➢ Figure S2. Observed (blue) and Modeled (red) daily maximum 8-hr average ozone in Beijing

[Figure]

➢
➢ Figure S3. Comparison between observed (blue) and modeled (red) monthly averaged diurnal variations of O₃ concentrations in the NCP (a), YRD (b) and PRD (c)
➢
➢ Table 4. Sensitivity of summer (July) daytime O3 to emission sectors for different regions (ppb)

| Sectors | NCP | YRD | PRD | India |
|---|---|---|---|---|
| Industry | 19.9 | 14.3 | 7.1 | 2.3 |
| Power | 6.1 | 7.0 | 4.9 | 2.7 |
| Residential | 4.1 | 1.9 | 1.6 | 3.3 |
| Transport | 8.9 | 9.2 | 5.9 | 10.0 |
| Biogenic | 28.7 | 28.9 | 12.0 | 7.6 |
| Fire | 1.4 | 0.8 | 0.3 | 0.1 |

[Figure]

➢

➢

➢ Fig. 8. Contributions to seasonally daily mean ozone (DJF, MAM, JJA, and SON) and MDA8 ozone (DJF M8, MAM M8, JJA M8, and SON M8) in Beijing (a), Shanghai (b), Guangzhou (c), and New Delhi (d) from different source regions.

Line 211: It is hard to figure out if the model can capture the site observations. A scatter plot is needed here.

➢ Reply: Following your suggestions, we have added a scatter plot in the revised manuscript: "Scatter plots of simulated and observed $O_3$ for four seasons suggest that model overestimates $O_3$ in most sites during winter, and exhibit better performance during summer (Fig. 3)."

[Figure]

Fig. 3. Scatter plot of simulated and observed seasonal mean ozone concentrations for Winter (a), Spring (b), Summer (c) and Fall (d).

Line 215: Figure S2 needs more explanation. How the OMI $NO_2$ product was used here? Criteria to filter the row anomaly? What is the solar zenith angle used to filter out data here? When computing the WRF-Chem $NO_2$ columns, how the model results are sampled? For instance, all the grids collocated with missing OMI data points due to cloud in OMI NO2 should not be used. How the OMI averaging kernel was used here? A detailed explanation is suggested.

➢ Reply: In the revised manuscript, we have added the following explanation: We evaluated also the spatial distribution of NO2 columns using the KNMI-DOMINO (Dutch OMI NO2) daily level-2 products of tropospheric NO2 column (www.temis.nl), with row anomaly removed (according to operational flagging), solar zenith angles less than 80⁰, and cloud fraction less than 0.2. The model results were sampled according to selected satellite data on a pair-to-pair basis. The matched model results were transformed by applying the OMI averaging kernel to the modeled vertical profiles of NO2 concentrations.

Line 217: In Figure S2, it is hard to conclude that the NO2 column from WRF-Chem is less than OMI NO2 column. A different plot or ratio plot is needed here. Also the emission deficiency may not be able to fully explain the high bias of nighttime ozone, because the

nighttime nitrogen chemistry may not be explicit in WRF-Chem. Again, I suggest to re-do the analysis using daytime ozone to eliminate the impacts from nighttime ozone.

➢ Reply: We have added difference plot in Figure S2. Model underpredicts $NO_2$ column in east China while overpredicts $NO_2$ column in central China.

➢ We agree that the emission deficiency cannot fully explain the high bias of nighttime ozone. As suggest by Fig. S3, In the NCP, model overestimates the titration effects, which can partly explain the underestimation of O3 in spring in the NCP (Fig. S3).

➢ To avoid confusion on how NO2 column affects nighttime ozone, we change the sentence to: "Fig. S2 indicates that modeled NO2 column values in east China are not as high as observed, but model overpredicts NO2 column in central China."

➢ Following your suggestions, we added analysis based on daytime ozone and MDA8 to eliminate the impacts from nighttime ozone in Table 4, Fig. 8, Fig. S9 and Fig. S10.

[Figure]

Fig. S2. Observed and WRF-Chem modeled $NO_2$ column and their differences (Model minus observation) for winter (a, e, i), spring (b, f, j), summer (c, g, k) and autumn (d, h, l)

Line 261: 'Lu et al., 2018b'?

➢ Reply: We have updated: summer monsoon systems (Lu et al., 2018b).

Line 287: This figure shows the Beijing-Tianjin-Hebei area is significantly influenced.

➢ Reply: Based on our definition in the DNSMI equation, the intensity of DNSMI represents the alternation of wind direction. In the DNSMI figure, BTH is located in the boundary between southerly winds and wind from northwest. As a result, the alternation intensity of wind vectors is high, but it is actually less influenced by southerly winds.

➢ To avoid this confusion, we added additional spatial plot of precipitation, which is another indication of summer monsoon (Fig. S7).

➢ We rephrased the sentence to: "North China is less influenced by the summer monsoon as suggested by the insignificant precipitation in summer (Fig. S7c). East China and South China are more affected as suggested by DNSMI values higher than 0.5 and more abundant precipitation (Fig. 5c and Fig. S7c)."

Line 292-293: The change of circulation is important because of the regional transport of air pollutants, but the change of cloudiness and precipitation pattern also plays an important role in the formation of ozone.

➢ Reply: Thanks for mentioning this important point. In our previous study (Lu et al., 2018), we found that summer monsoon can bring about cloudy and rainy weather conditions (Fig. S7, removement of ozone precursors), weaker solar radiation, and lower temperature (Lu et al., 2018b). The onset of the summer monsoon is also associated with strong air convergence and uplift, which is not favorable for the accumulation of O3 and its precursors (Lu et al, 2018b).

➢ We have added this information in the revised manuscript.

➢ Lu, X., Zhang, L., Liu, X., Gao, M., Zhao, Y. and Shao, J.: Lower tropospheric ozone over India and its linkage to the South Asian monsoon, Atmos. Chem. Phys., 18(5), 3101–3118, doi:10.5194/acp-18-3101-2018, 2018b.

Line 311: As discussed above, a paragraph is needed here to explain how the sensitivity was calculated.

➢ Reply: In the revised manuscript, we have added the following sentence to explain how sensitivity was calculated:

➢ The sensitivity is defined as the responses of $O_3$ concentration to the elimination of each source sector ($O_{3\,with\,all\,emissions} - O_{3\,without\,each\,sector}$).

Line 314: I disagree with this statement. From Fig 5, looks like emissions from Transportation sector contribute more than the Industry sector in China.

➢ Reply: To avoid confusions like this, we added Table 3 in the manuscript to better illustrate the relative importance.

➤ As shown in Table 3, emissions from transportation sector contribute to 2.8, 4.0 and 3.6 in summer for NCP, YRD and PRD, lower than industrial sector (8.3, 8.3 and 4.7).

Line 317: Li et al. (2017) is missing.

➤ Reply: We have added it in the revised manuscript.
➤ Li, G., Bei, N., Cao, J., Wu, J., Long, X., Feng, T., Dai, W., Liu, S., Zhang, Q., and Tie, X.: Widespread and persistent ozone pollution in eastern China during the non-winter season of 2015: observations and source attributions, Atmos. Chem. Phys., 17, 2759–2774, https://doi.org/10.5194/acp-17-2759-2017, 2017.

Line 341: Is 'biomass burning' here equivalent to 'Fire' sector in Figure 5?

➤ Reply: Yes, to avoid confusion, we added explanation in Figure 5.

Line 347: Looks like the biogenic emissions dominate the ozone production in India.

➤ Reply: Biogenic emissions are the major source for emissions of VOCs. The biogenic emitted amount is close to those from all anthropogenic sectors in China and greater than anthropogenic sources in India.
➤ As a result, the zero-out method would show that biogenic emissions dominate over other emission sectors (Table S2).
➤ However, due to the non-linearity of ozone chemistry, it is not reasonable to conclude that biogenic emissions dominate the contribution as biogenic VOC react with anthropogenic NOx. From Fig. 5, we can see that transport emissions also play important roles. In some regions like India, the values are higher than biogenic emissions (Table 3).
➤ As biogenic emissions are difficult to control, in the discussion we highlight the relative importance of anthropogenic emission sectors.

Table S2 Amount of emission species from anthropogenic, biogenic and biomass burning sources

| China | $SO_2$ | $NO_x$ | NMVOC | $NH_3$ | CO | $PM_{2.5}$ | BC | OC |
|---|---|---|---|---|---|---|---|---|
| Anthropogenic | 28.5 | 29.2 | 28.1 | 10.7 | 180.2 | 11.9 | 1.8 | 3.2 |
| Biogenic | | 0.3 | 24 | | 2.9 | | | |
| Biomass burning | 0.2 | 0.2 | 0.6 | 0.1 | 6.7 | 0.6 | 0.04 | 0.3 |
| India | | | | | | | | |
| Anthropogenic | 8.4 | 8.9 | 16.0 | 9.4 | 61.8 | 4.9 | 1.0 | 2.5 |
| Biogenic | | 0.3 | 18.7 | | 2.7 | | | |

| Biomass burning | 0.1 | 0.7 | 0.8 | 0.2 | 18.5 | 1.9 | 0.1 | 0.9 |

Table 3. Sensitivity of seasonal $O_3$ to emission sectors for different regions (ppb)

| Sectors | Seasons | NCP | YRD | PRD | India |
|---|---|---|---|---|---|
| Industry | Winter | -4.1 | -1.5 | 4.5 | 2.1 |
| | Spring | -0.3 | 3.8 | 6.5 | 1.7 |
| | Summer | 8.3 | 8.3 | 4.7 | 1.6 |
| | Autumn | -1.4 | 1.7 | 7.1 | 2.1 |
| Power | Winter | -5.6 | -7.5 | -1.2 | 1.7 |
| | Spring | -3.2 | -2.2 | 2.2 | 2.3 |
| | Summer | 2.7 | 2.9 | 3.3 | 1.9 |
| | Autumn | -3.3 | -3.3 | 2.1 | 2.4 |
| Residential | Winter | 5.1 | 7.7 | 6.6 | 4.2 |
| | Spring | 2.4 | 2.5 | 1.9 | 2.4 |
| | Summer | 2.5 | 1.4 | 1.1 | 2.2 |
| | Autumn | 2.2 | 2.2 | 1.6 | 3.2 |
| Transport | Winter | -8.5 | -8.0 | 0.2 | 7.6 |
| | Spring | -3.7 | -1.5 | 3.4 | 7.9 |
| | Summer | 2.8 | 4.0 | 3.6 | 6.7 |
| | Autumn | -4.3 | -3.3 | 3.0 | 8.9 |
| Biogenic | Winter | 0.3 | 1.0 | 3.8 | 4.8 |
| | Spring | 4.3 | 6.6 | 7.5 | 5.6 |
| | Summer | 19.2 | 18.5 | 9.4 | 5.7 |
| | Autumn | 5.7 | 6.5 | 11.4 | 8.0 |
| Fire | Winter | 0.1 | 0.2 | 2.3 | 0.6 |
| | Spring | 1.1 | 1.8 | 2.6 | 1.1 |
| | Summer | 3.8 | 4.0 | 1.2 | 0.2 |
| | Autumn | 1.2 | 1.4 | 1.9 | 0.5 |

Line 352: I disagree with this statement. Due to the VOC-limited ozone production, biogenic emissions has the largest contribution (Figure 5S).

➤ Reply: We agree with you that biogenic emissions show the largest contribution from Figure 5S.

➤ Due to the non-linearity of ozone chemistry, it is not reasonable to conclude that biogenic emissions are important than anthropogenic emissions as biogenic VOC react with anthropogenic NOx.

➤ As biogenic emissions are not easy to control, here we would like to identify important anthropogenic sectors in terms of ozone control.

➤ To avoid confusion, we rephrase the sentence to: "Our results highlight the importance of industrial sources and biogenic emissions in O$_3$ formation in east China, consistent with the conclusions of Li et al. (2017)."

Line 771 Table 2: Does the 'Outside' stand for outside of China? Need clarification.

➤ Reply: We have added clarification: Outside sources represent sources outside China for the discussed three regions in China, and sources outside India for India, including also transport from upper boundary of the model.

Line 779 Figure 1: It is very hard to tell the Purple sold circle in the map. I suggest using a different shape such as solid star or triangle here.

➤ Reply: We have updated with solid triangle here.

[Figure]

Line 783 Figure 2: As mentioned above, it is hard to see the model performance. A scatter plot is suggested here.

➢ Reply: We have added a scatter plot in the revised manuscript: "Scatter plots of simulated and observed O3 for four seasons suggest that model overestimates O3 in most sites during winter, and exhibit better performance during summer (Fig. 3)."

[Figure]

Fig. 3. Scatter plot of simulated and observed seasonal mean ozone concentrations for Winter (a), Spring (b), Summer (c) and Fall (d).

Line 787 Figure 3: The scale in y-axis is improper. Please re-plot the figure with y-axis from 0 to 80 ppbv or 100 ppbv. Also the current plot shows little difference between these two lines. A table shows some statistics such as NMB and RMSE is suggested.

➢ Reply: We have changed the scale in y-axis to 100 ppbv. In the revised manuscript, we also added a table to show statistics of model evaluation, mean bias, root square error, normalized mean bias, normalized mean error and correlation coefficient.

➢ We added also description in the context: "Detailed model evaluation statistics are documented in Table 2."

Table 2 Model evaluation statistics

| Regions | NCP | YRD | PRD | India |
|---|---|---|---|---|
| Mean Bias | -3.8 | -1.8 | 3.1 | -2.0 |
| Root Mean Square Error | 6.4 | 5.5 | 7.9 | 4.4 |
| Normalized Mean Bias | -13.3% | -6.2% | 10.7% | -5.6% |
| Normalized Mean Error | 18.7% | 14.9% | 21.2% | 11.1% |
| R | 0.98 | 0.96 | 0.84 | 0.91 |

Line 797 Figure 4: what is the 'near surface wind field' here? 10 m or 850 hPa wind?

➢ Reply: 10m. We have added this information in the caption of Figure 4.

**Referee #2**

Review of the manuscript titled "Ozone Pollution over China and India: Seasonality and Sources." (By Meng Gao et al.)

The paper presents the results of an WRF-Chem modeling and analyzed the seasonality of O₃ and its sources in both China and India. They derived the modeling results from the sensitivity tests (authors named this as 'a factor separation approach') and they explained that the importance of industrial sources in summer in China and the transport vehicle sector in all seasons in India. Also authors described the contributions from other regions. The content of the manuscript is reasonable and the results also seem convincing. However, all the discussion is mostly based on modeling results, and thus the evaluation and validation of WRF-Chem results has to be made more rigorously. Also some of the descriptions in the text should be changed to be in more quantitative way, and authors can provide a statistical and quantitative modeling (or observational) results on the relations between O₃ and its precursors.

➢ Reply: Thank you for the careful reading and valuable comments.
➢ In the revised manuscript, we have added more model evaluations, including evaluations of MDA8 as suggested by Reviewer 1 and evaluation of diurnal distributions of O₃ in China.

[Figure]

Figure S2. Observed (blue) and Modeled (red) daily maximum 8-hr average ozone in Beijing

[Figure]

Figure S3. Comparison between observed (blue) and modeled (red) monthly averaged diurnal variations of $O_3$ concentrations in the NCP (a), YRD (b) and PRD (c)

➢ We also added Table 2 to quantitatively show how model performs, and Table 3 and Table 4 to quantitatively display the associations between seasonal $O_3$ concentrations and different emission sectors for different regions.

(Major Comments)

1) Line 166, 23 source regions: In most figures, regions are divided into four areas (NCP, YRD, PRD, and India) for current analysis. However, in Fig. S1, authors provided only province borders in China. Please redraw the boundaries of four regions (and 8 additional regions, as employed in Fig. 6) in the Figures.

➢ Reply: Four regions are marked in Fig. S1(b); For 8 additional regions, NW China, NW China, SW China, N India, and S India are marked in Fig. S1(a); Central China, SE China and South China are marked in Fig. S1(b).

[Figure]

2) In Fig. 2, in winter, for example, it is considered that the $O_3$ concentration level in East China shows considerable biases. It is suggested that at least the statistical results of four regions (NCP, YRD, PRD, and India) should be explored. Statistical results such as RMSE, IOA, etc (together with correlation coefficients as indicated in Line 228) should be calculated (or described in the manuscript) for at least this four regions.

➢ Reply: In the revised manuscript, we added statistical results for four regions in Table 2, including mean bias, root square error, normalized mean bias, normalized mean error and correlation coefficient.

Table 2 Model evaluation statistics

| Regions | NCP | YRD | PRD | India |
|---|---|---|---|---|
| Mean Bias | -3.8 | -1.8 | 3.1 | -2.0 |
| Root Mean Square Error | 6.4 | 5.5 | 7.9 | 4.4 |

| | | | | |
|---|---|---|---|---|
| Normalized Mean Bias | -13.3% | -6.2% | 10.7% | -5.6% |
| Normalized Mean Error | 18.7% | 14.9% | 21.2% | 11.1% |
| R | 0.98 | 0.96 | 0.84 | 0.91 |

3) In Fig. 3, if the y-axis scale is lowered to 80 ppb (or even lower), the differences (between modeling and observations) is unlikely to be ignored. It appears that the spring season (Mar. Apr, May) in NCP is so low that the model result looks nearly doubled, and also 3 months (Oct Noc Dec) in PRD show considerable bias. India may drive higher $O_3$ measurement in April, but the model is not simulated to be as high as measurement. Explain the reasons why the monthly measured average is high in India but why not simulated in April, through WRF-Chem model.

> Reply: This comment is consistent with the problem that raised by Referee 1. Following the suggestions, we lower the y axis to 100 ppb.
> We agree that the model underestimates ozone in NCP in spring. We acknowledged this problem in the paper: The seasonality of observed O3 concentrations is reproduced well in these four regions (Fig. 4), although concentrations are underestimated in the NCP in spring.
> In the revised manuscript, we added sentences to acknowledge the model biases in PRD: $O_3$ concentrations in October, November and December in the PRD region are overestimated by the model.
> For India, although the countrywide mean concentrations are close to the model estimate, spatial model evaluation results suggest that ozone concentrations are underestimated, particularly in Jabalpur in Central India (Fig. 2). In other sites, model show better consistency, but the mismatch in Jabalpur lead to the low bias.
> The large underestimation in Jabalpur was also found in other studies (Table S2 and S3 in Sharma et al., 2017). The large bias is attributed to the strong spatial heterogeneity in land types within a small area (Sharma et al., 2017).
> In the revised manuscript, we added the following explanation: "The simulated magnitudes of O3 in India are generally consistent with observations, though lower in central India and in May. The high concentrations of O3 in India were not captured by the model is mainly because of the large underestimation in Jabalpur (Central India) with complex terrain. Model's coarse resolution and poor capability of resolving strong spatial heterogeneity in land types within a small area have led to this mismatch, which was also found in Sharma et al. (2017). "
> Sharma, A., Ojha, N., Pozzer, A., Mar, K. A., Beig, G., Lelieveld, J., and Gunthe, S. S.: WRF-Chem simulated surface ozone over south Asia during the pre-monsoon: effects of emission inventories and chemical mechanisms, Atmos. Chem. Phys., 17, 14393–14413, https://doi.org/10.5194/acp-17-14393-2017, 2017.

[Figure]

4) In this study, as a Monsoon Index, only wind field index was chosen (Li and Zeng, 2002). Of course, we know that the wind field is an important factor that causes precipitation by the Monsoon cycle, but the distribution of precipitation itself is one of the most important controlling factors in dealing with $O_3$. Collect reanalyzed precipitation data (i.e., GPCP data) and plot horizontal distributions in supplementary material, and analyze both precipitations vs. $O_3$ modeling patterns to explore the regional characteristics more directly.

> ➢ Reply: Following your suggestion, we added spatial distribution of GPCP precipitation in the supplementary material (Fig. S7).
> ➢ In the revised paper, we added the following sentences to describe precipitations vs. O3 modeling patterns:
> ➢ Fig. S7(c) also indicates that most areas of India and South China are influenced by summer monsoon.
> ➢ The alternation of wind vectors and precipitation (Fig. S7) from winter to summer results also in changes in upwind areas and abundance of O3 precursors, modulating the severity of O3 pollution.
> ➢ Besides, cloudy and rainy conditions (Fig. S7) associated with the summer monsoon are not conducive to photochemical production of O3 (Tang et al., 2013).
> ➢ Summer monsoon can bring about cloudy and rainy weather conditions (Fig. S7, removement of ozone precursors), weaker solar radiation, and lower temperature (Lu et al., 2018b). The onset of the summer monsoon is also associated with strong air convergence and uplift, which is not favorable for the accumulation of O3 and its precursors (Lu et al, 2018b).

[Figure]

Fig. S7. Mean precipitation (mm/day) in DJF (a), MAM (b), JJA (c) and SON (d) inferred from the Global Precipitation Climatology Project (GPCP) version 2.3 dataset

5) It seems reasonable to analyze O₃ during the night and day separately, to see the over- (or under-) estimation of titration effects, day and night separately, as referee 1 pointed out.

➤ Reply: To see if the model over- or under-estimate the titration effects, we added comparison between the observed (blue) and modeled (red) monthly averaged diurnal variations of O3 concentrations in the NCP (a), YRD (b) and PRD (c)

➤ In the NCP, model overestimates the titration effects, which can partly explain the underestimation of O₃ in spring in the NCP (Fig. S3). The titration effects are also overestimated in the YRD, but underestimated in the PRD in winter (Fig. S3).

➤ We added results based on daytime ozone: "The negative sensitivity of O3 to the transport and power sectors may be also caused by the nighttime titration effects. In winter, daytime (12:00-18:00 local time) mean O3 exhibit also negative sensitivity to transportation sector, and similar distribution with daily mean O3 sensitivity (Fig. S9a and S9b), suggesting nighttime titration effects might not be the major reason in winter. However, daytime mean and daily mean O3 exhibit different patterns of sensitivity to transportation sector in highly urbanized regions in summer, which could be related to nighttime titration effects. As indicated in Fig. S10, O3 sensitivity to transportation sector in Beijing is positive during the day but negative during the night. " (ref. Fig. S9, Fig. S10, Table 4)

➢ We also compared the results of region source contribution to daily mean ozone and MDA8 separately, and found the conclusions are generally consistent, except that local sources exhibit slightly more important roles in the formation of MDA8 of ozone concentrations. (ref. Fig. 7)

[Figure]

Figure S3. Comparison between observed (blue) and modeled (red) monthly averaged diurnal variations of O₃ concentrations in the NCP (a), YRD (b) and PRD (c)

[Figure]

Fig. 7. Contributions of source regions to seasonally mean O₃ and MDA8 in Beijing (a), Shanghai (b), Guangzhou (c), and New Delhi (d)

6) Add quantitative emissions for gas such as $SO_2$ $NO_x$, $NH_3$, and VOCs together with particulate matters for the whole 2013 year. Also, describe quantitatively the total of biogenic emission and biomass burning emission over both China and India for 2013 year, and compare to other emissions (i.e., $SO_2$, NOx, NH3, and VOCs)

> ➢ Reply: Following your suggestion, in the revised manuscript, we added Table S2 to list the quantitative emissions of anthropogenic, biogenic and biomass burning for both China and India.
> ➢ We added description in the manuscript: The large sensitivity of O3 to biogenic emissions is associated with the massive emissions of biogenic VOCs (Table S2). The amount of biogenic VOCs is comparable to those emitted from all anthropogenic sectors in China, and greater than anthropogenic VOCs in India (Table S2).

Table S2 Amount of emission species from anthropogenic, biogenic and biomass burning sources

| China | $SO_2$ | $NO_x$ | NMVOC | $NH_3$ | CO | $PM_{2.5}$ | BC | OC |
|---|---|---|---|---|---|---|---|---|
| Anthropogenic | 28.5 | 29.2 | 28.1 | 10.7 | 180.2 | 11.9 | 1.8 | 3.2 |
| Biogenic | | 0.3 | 24 | | 2.9 | | | |
| Biomass burning | 0.2 | 0.2 | 0.6 | 0.1 | 6.7 | 0.6 | 0.04 | 0.3 |
| India | | | | | | | | |
| Anthropogenic | 8.4 | 8.9 | 16.0 | 9.4 | 61.8 | 4.9 | 1.0 | 2.5 |
| Biogenic | | 0.3 | 18.7 | | 2.7 | | | |
| Biomass burning | 0.1 | 0.7 | 0.8 | 0.2 | 18.5 | 1.9 | 0.1 | 0.9 |

(Minor Comments)

1) Line 133: Year 2010 isn't so old ? The changes in emissions should be mentioned briefly in the revised manuscript.

> ➢ Reply: The emissions we used were updated to year 2012 (not 2010) for China, as mentioned in the manuscript "In this study, the emissions in China were updated with the MEIC (Multi-resolution Emission Inventory for China, http://www.meicmodel.org/) inventory for year 2012".
> ➢ For India, there is less information and updated emission inventory available.
> ➢ In the revised manuscript, we added a sentence to describe the changes in emissions from 2012 to 2013: "From 2012 to 2013, emissions of $SO_2$ and $NO_x$ in China declined by 11% and 5%, while emissions of other species did not exhibit a significant change (Zheng et al., 2018)".
> ➢

2) Line 245 Fig. S3 -> Fig. S4?

> Reply: Yes, thanks for the correction. We have modified it in the revised manuscript.

3) Line 246, satellite NO2 column (Fig. S2) indicates only day time? If this is the case, in Fig. S2, modeling results also should be only day-time results?

> Reply: Yes, the modeling results were compared against observations on a pair-to-pair basis. Modeling results are only day-time results.

> In the revised manuscript, we have added more information about how model results were compared against satellite data, including averaging kernel, etc.

> "We evaluated also the spatial distribution of NO2 columns using the KNMI-DOMINO (Dutch OMI NO2) daily level-2 products of tropospheric NO2 column (www.temis.nl), with row anomaly removed (according to operational flagging), solar zenith angles less than $80^0$, and cloud fraction less than 0.2. The model results were sampled according to selected satellite data on a pair-to-pair basis. The matched model results were transformed by applying the OMI averaging kernel to the modeled vertical profiles of NO2 concentrations."

4) Do not you have any VOC observations?

> Reply: VOCs measurements have not been regularly observed and released by the Chinese National Environmental Modeling Center (CNEMC). Previous model evaluation during a field campaign in summer in Shanghai suggest that the WRF-Chem slightly underestimated observed concentrations of VOCs (Tie et al., 2013).

> The model performances can differ with time and regions, but it has not been well explored due the lack of and inaccessibility to VOC observations.

> Recently, more efforts have been made to measure VOC in Guangdong and Hong Kong. With more measurements becoming available, we will explore it in future studies.

> Tie, X., Geng, F., Guenther, A., Cao, J., Greenberg, J., Zhang, R., Apel, E., Li, G., Weinheimer, A., Chen, J., and Cai, C.: Megacity impacts on regional ozone formation: observations and WRF-Chem modeling for the MIRAGE-Shanghai field campaign, Atmos. Chem. Phys., 13, 5655–5669, https://doi.org/10.5194/acp-13-5655-2013, 2013.